**Technical Report**

# Parallel sequencing of extrachromosomal circular DNAs and transcriptomes in single cancer cells

Rocío Chamorro González[1,2], Thomas Conrad [3], Maja C. Stöber [4,5,6], Robin Xu [1,2], Mădălina Giurgiu [1,2,7], Elias Rodriguez-Fos[1,2], Katharina Kasack[8], Lotte Brückner [2,9], Eric van Leen [1,2], Konstantin Helmsauer [1,2], Heathcliff Dorado Garcia [1,2], Maria E. Stefanova [10,11], King L. Hung [12], Yi Bei [1,2], Karin Schmelz[1], Marco Lodrini[1], Stefan Mundlos[10,11,13], Howard Y. Chang [12,14], Hedwig E. Deubzer [1,2,15,16], Sascha Sauer [4], Angelika Eggert [1,15,16], Johannes H. Schulte [1,15,16], Roland F. Schwarz[4,17,18], Kerstin Haase [1,2,15], Richard P. Koche [19,20] & Anton G. Henssen [1,2,9,15,20] ✉

Extrachromosomal DNAs (ecDNAs) are common in cancer, but many questions about their origin, structural dynamics and impact on intratumor heterogeneity are still unresolved. Here we describe single-cell extrachromosomal circular DNA and transcriptome sequencing (scEC&T-seq), a method for parallel sequencing of circular DNAs and full-length mRNA from single cells. By applying scEC&T-seq to cancer cells, we describe intercellular differences in ecDNA content while investigating their structural heterogeneity and transcriptional impact. Oncogene-containing ecDNAs were clonally present in cancer cells and drove intercellular oncogene expression differences. In contrast, other small circular DNAs were exclusive to individual cells, indicating differences in their selection and propagation. Intercellular differences in ecDNA structure pointed to circular recombination as a mechanism of ecDNA evolution. These results demonstrate scEC&T-seq as an approach to systematically characterize both small and large circular DNA in cancer cells, which will facilitate the analysis of these DNA elements in cancer and beyond.

Measuring multiple parameters in the same cells is key to accurately understand biological systems and their changes during diseases[1]. In the case of circular DNAs, it is critical to integrate DNA sequence information with transcriptional output measurements to assess their functional impact on cells. At least three types of circular DNAs can be distinguished in human cells[2–5]: (1) small circular DNAs (<100 kb)[6], which have been described under different names including eccDNAs[6], microDNAs[4], apoptotic circular DNAs[6], small polydispersed circular DNAs[7] and telomeric circular DNAs or C-circles[8]; (2) T cell receptor excision circles (TRECs)[9]; and (3) large (>100 kb), oncogenic, copy number-amplified circular extrachromosomal DNAs[10,11] (referred to as ecDNA and visible as double minute chromosomes during metaphase[12]). Despite our increasing ability to characterize multiple features in single cells[13], an in-depth characterization of circular DNA content, structure and sequence in single cells remains elusive with current approaches.

In cancer, oncogene amplifications on ecDNA are of particular interest because they potently drive intercellular copy number heterogeneity through their unique ability to be replicated and unequally segregated during mitosis[14–19]. This heterogeneity enables tumors to

adapt and evade therapies[2,20–22]. Indeed, patients with ecDNA-harboring cancers have adverse clinical outcomes[11]. Recent investigations indicate that enhancer-containing ecDNAs interact with each other in nuclear hubs[17,23] and can influence distant chromosomal locations in *trans*[23,24]. This suggests that even ecDNAs not harboring oncogenes may be functional[23,24]. Furthermore, we recently revealed that tumors harbor an unanticipated repertoire of smaller, copy number-neutral circular DNAs of yet unknown functional relevance[3].

In this study, we report single-cell extrachromosomal circular DNA and transcriptome sequencing (scEC&T-seq), a method that enables parallel sequencing of all circular DNA types, independent of their size, content and copy number, and full-length mRNA in single cells. We demonstrate its utility for profiling single cancer cells containing both structurally complex multifragmented ecDNAs and small circular DNAs.

## Results

### scEC&T-seq detects circular DNA and mRNA in single cells

Current state-of-the-art circular DNA purification approaches involve three sequential steps, that is, isolation of DNA followed by removal of linear DNA through exonuclease digestion and enrichment of circular DNA by rolling circle amplification[3,6,25]. We reasoned that this approach may be scaled down to single cells and when combined with Smart-seq2 (ref. 26) may allow the parallel sequencing of circular DNA and mRNA. To benchmark our method in single cells, we used neuroblastoma cancer cell lines, which we had previously characterized in bulk populations[3]. We used FACS to separate cells into 96-well plates (Fig. 1a, Supplementary Fig. 1a,b and Supplementary Table 1). DNA was separated from polyadenylated RNA, which was captured on magnetic beads coupled to single-stranded sequences of deoxythymidine (Oligo dT) primers, similarly to previous approaches[27]. DNA was subjected to exonuclease digestion, as successfully performed in bulk cell populations in the past, to enrich for circular DNA[3,6,25] (Fig. 1b). DNA subjected to PmeI endonuclease before exonuclease digestion served as a negative control[3]. In a subset of cases, DNA was left undigested as an additional control (Fig. 1b). The DNA remaining after the different digestion regimens was amplified. The amplified DNA was subjected to Illumina paired-end sequencing and in some cases to long-read Nanopore sequencing (Fig. 1a). The sequence composition of circular DNAs was analyzed and genomic origin was inferred in circularized regions using previously established computational algorithms for circular DNA analysis[3].

To evaluate the performance of our scEC&T-seq method, we first assessed mitochondrial DNA (mtDNA) detection and enrichment because mtDNA is present in all cells, is digested by PmeI and, due to its circularity and extrachromosomal nature, serves as a positive control. A significantly higher percentage of reads mapping to mtDNA was detected after longer exposure of the DNA of single cells to exonuclease ($P < 2.2 \times 10^{-16}$, two-sided Welch's $t$-test; Fig. 1c,d and Supplementary Fig. 1c,d). This was also the case for all other circular DNA elements ($P < 2.2 \times 10^{-16}$, two-sided Welch's $t$-test; Fig. 1e), indicating significant enrichment of circular DNA. Significant enrichment of ecDNA regions, that is, large (>100 kb) circular DNAs containing oncogenes, was observed after 1-day exonuclease digestion ($P = 2.10 \times 10^{-5}$, two-sided Welch's $t$-test; Supplementary Fig. 1e). This enrichment was not as pronounced as that of smaller circular DNAs after prolonged 5-day exonuclease digestion, suggesting that ecDNA may be less stable in the presence of exonuclease compared to smaller circular DNAs, or that small circular DNAs are more efficiently amplified by φ29 polymerase (Supplementary Fig. 1e,f). PmeI endonuclease incubation before 5-day exonuclease digestion significantly reduced reads mapping to mtDNA by 404.8 fold ($P < 2.2 \times 10^{-16}$, two-sided Welch's $t$-test; Fig. 1c,d and Supplementary Fig. 1c). Similar depletion was observed for reads mapping to circular DNAs containing PmeI recognition sites, confirming specific enrichment of circular DNA through our scEC&T-seq protocol ($P < 2.2 \times 10^{-16}$,

two-sided Welch's $t$-test; Fig. 1f and Supplementary Fig. 1g,h). Significant concordance between Illumina- and Nanopore-based detection of circular DNAs suggested reproducible detection independent of sequencing technology (two-sided Pearson correlation, $R = 0.95$, $P < 2.2 \times 10^{-16}$; Supplementary Fig. 2a–d). Thus, scEC&T-seq enables the isolation and sequencing of circular DNAs from single cells.

The separated mRNA from the same cells was processed using Smart-seq2 (ref. 26,27) (Fig. 1a and Supplementary Note 1). We detected on average 9,058 ± 1,163 (mean ± s.d.) full mRNA transcripts from different genes per cell (Supplementary Fig. 3a–c and Supplementary Table 2). Unsupervised clustering separated both cell line populations (Supplementary Fig. 3d,e). To test whether scEC&T-seq provided high-quality mRNA sequencing data, we assessed cell cycle signature gene expression and classified single cells into three cell cycle phases (G1, S, G2/M; Supplementary Fig 3f). The cell cycle distributions inferred from scEC&T-seq matched those measured using FACS-based cell cycle analysis, confirming its accuracy (Supplementary Fig. 3g). Thus, scEC&T-seq not only enables the enrichment and detection of circular DNAs, but also allows parallel measurement of high-quality, full transcript mRNA in single cancer cells.

### scEC&T-seq detects recurrent ecDNAs in single cells

Only circular DNAs conferring a fitness advantage are expected to be clonally present in a cancer cell population[22]. We recently found that tumors on average harbor more than 1,000 individual circular DNAs, most of which are small (<100 kb), lack oncogenes and do not contribute to oncogene amplification[3]. Their intercellular differences, however, remain unexplored and it is still unclear whether small circular DNAs can confer a fitness advantage and are clonally propagated in cancer cells[10]. Consistent with our previous reports in bulk populations[3], the average number of individual circular DNA regions identified using scEC&T-seq varied between 97 and 1,939 (median = 702) per single cell in neuroblastoma cell lines (Fig. 2a). The circular DNA size distribution and genomic origin was similar between single cells and mirrored the distribution observed in bulk sequencing[3] (minimum = 30 bp, maximum = 1.2 Mb, median = 21,483 kb; Fig. 2a and Supplementary Fig. 4a,b). All analyzed cells were alive at the time of sorting (Supplementary Fig. 1a,b) and most (>95%) circular DNAs detected in single cells were larger than apoptotic circular DNAs, suggesting that most circular DNAs were not a result of apoptosis, as suggested by other reports[6] (Fig. 2a and Supplementary Fig. 4a). Thus, each cancer cell contains a wide spectrum of individual circular DNAs from different genomic contexts.

As expected, most small circular DNAs did not harbor oncogenes[10]. The overall proportion of small circular DNAs detected recurrently in cells was low (Fig. 2b–d and Supplementary Fig. 4c). This indicates that only a small subset of small circular DNAs is clonally propagated in cancer cells. In line with their known oncogenic role in cancer and the positive selective advantages, amplified, oncogene-containing ecDNAs were recurrently detected in cells (Fig. 2b–d), which was validated by FISH (Fig. 2b and Supplementary Fig. 5a–c). Even though the functional relevance of small circular DNAs cannot be excluded, the observed high subclonality suggests that they do not contribute to cancer cell fitness to the same extent as clonal oncogene-amplifying ecDNA.

### Complex multifragmented ecDNAs are detectable in single cells

We and others recently showed that ecDNAs are complex structures, sometimes containing rearranged fragments from different chromosomes[23,28–30]. Considering that scEC&T-seq was able to recurrently detect megabase-sized ecDNAs harboring the oncogenes *MYCN*, *CDK4* or *MDM2* (Fig. 2b), we asked whether scEC&T-seq could provide insights into ecDNA structures. Indeed, scEC&T-seq captured multifragment ecDNAs in almost all single cells recapitulating the

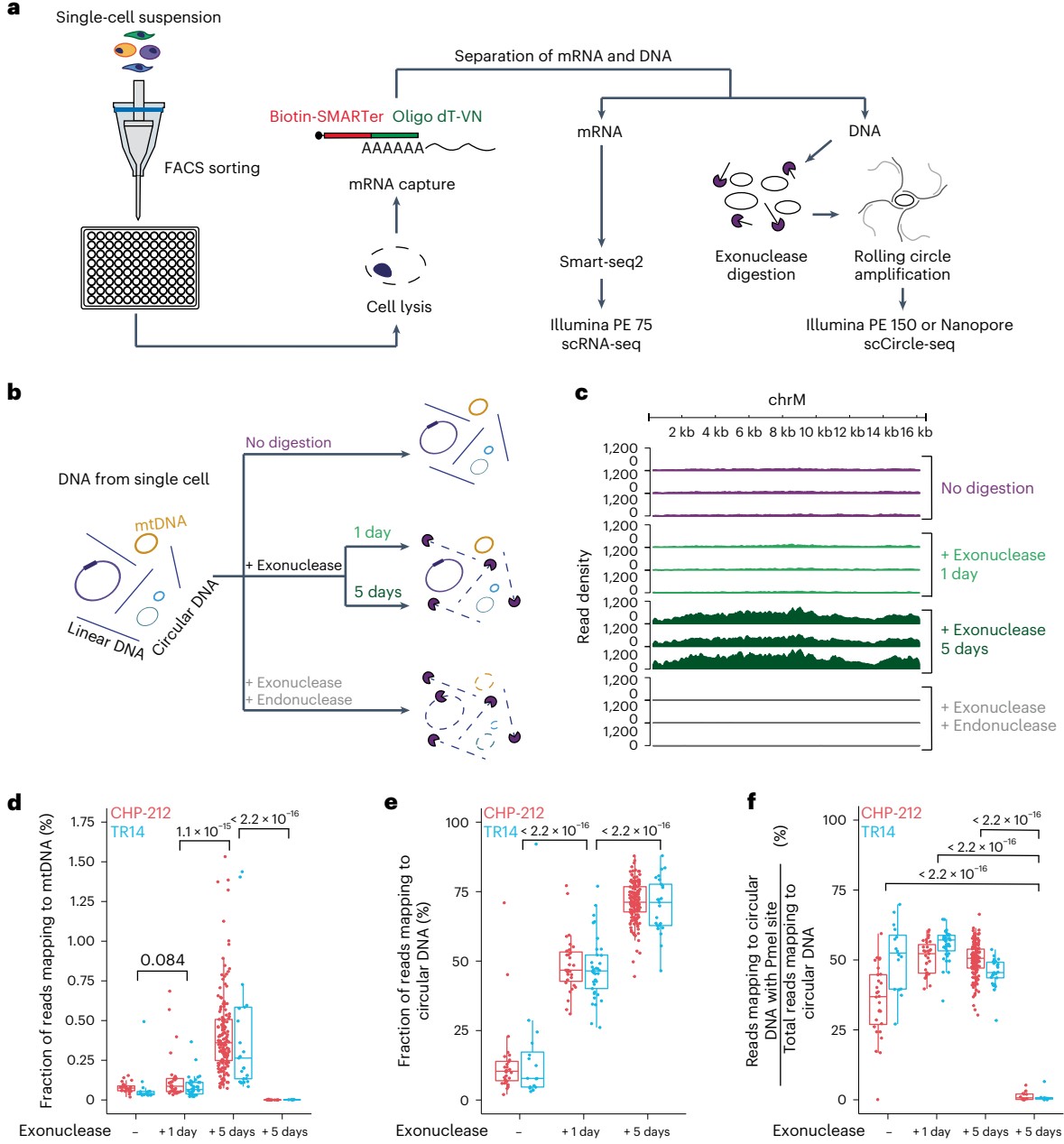

**Fig. 1 | scEC&T-seq enables enrichment and detection of circular DNA in single cells. a**, Schematic of the scEC&T-seq method. **b**, Schematic representation of the experimental conditions and expected outcomes. **c**, Genome tracks comparing read densities on mtDNA (chrM) in three exemplary CHP-212 cells for each experimental condition tested. Top to bottom, No digestion (purple), 1-day exonuclease digestion (light green), 5-day exonuclease digestion (dark green) and endonuclease digestion with PmeI before 5-day exonuclease digestion (gray). **d**, Fraction of sequencing reads mapping to mtDNA in each experimental condition in CHP-212 (red) and TR14 (blue) cells. **e**, Fraction of sequencing reads mapping to circular DNA regions identified by scEC&T-seq in each experimental condition in CHP-212 and TR14 cells. **f**, Fraction of sequencing reads mapping to circular DNA regions with the endonuclease PmeI targeting the sequence identified by scEC&T-seq in each experimental condition in CHP-212 and TR14 cells. **d**–**f**, Sample size is identical across conditions: no digestion ($n = 16$ TR14 cells, $n = 28$ CHP-212 cells); 1-day exonuclease digestion ($n = 37$ TR14 cells, $n = 31$ CHP-212 cells); 5-day exonuclease digestion ($n = 25$ TR14 cells, $n = 150$ CHP-212 cells); and endonuclease digestion with PmeI before 5-day exonuclease digestion ($n = 6$ TR14 cells, $n = 12$ CHP-212 cells). All statistical analyses correspond to a two-sided Welch's $t$-test. $P$ values are shown. In all boxplots, the boxes represent the 25th and 75th percentiles with the center bar as the median value and the whiskers representing the furthest outlier $\leq 1.5\times$ the interquartile range (IQR) from the box.

previously described element structures found in bulk populations[23,28] (Fig. 3a,b). At least one variant-supporting read per ecDNA breakpoint was detectable in approximately 30% of single cells (Supplementary Table 3). Further quantification of ecDNA junction-spanning reads and computational structural variant (SV) detection both from short- and long-read sequencing confirmed the interconnectedness of segments

(Supplementary Fig. 6a–p and Supplementary Tables 4 and 5). Such SVs can lead to fusion transcript expression on ecDNA[3]. Indeed, fusion transcripts could be identified in single cells using scEC&T-seq (Fig. 3c and Supplementary Fig. 7). Thus, scEC&T-seq is sufficiently sensitive to detect ecDNA-associated SVs and resulting fusion gene expression in single cells.

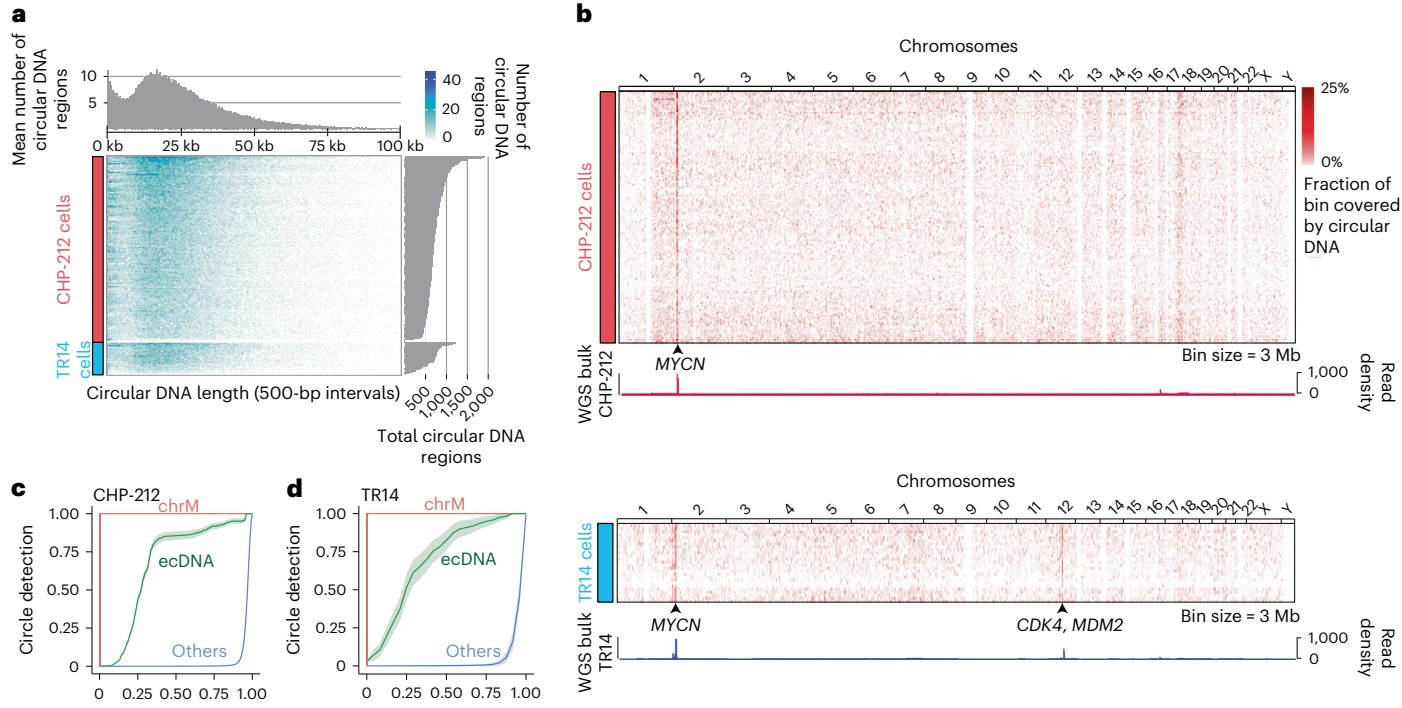

**Fig. 2 | Oncogene-containing ecDNAs are recurrently identified in neuroblastoma single cells. a**, Heatmap displaying the number and length of individual circular DNA regions (<100 kb) identified by scEC&T-seq in CHP-212 and TR14 neuroblastoma single cells (n = 150 CHP-212 cells, n = 25 TR14 cells; bin size = 500 bp) with density distribution for circular DNA sizes (top) and overall circular DNA counts (right). **b**, Heatmap of genome-wide circular DNA density in CHP-212 and TR14 neuroblastoma single cells (top: n = 150 CHP-212 cells, bin size = 3 Mb; bottom: n = 25 TR14 cells, bin size = 3 Mb), and genome tracks displaying genome-wide read density from WGS in bulk cell populations. The location of the *MYCN* gene in chromosome 2 is shown. **c**, **d**, Recurrence analysis in CHP-212 (n = 150) (**c**) and TR14 (n = 25) (**d**) cells displayed as the fraction of cells containing a detected circular DNA from each circular DNA type. ecDNA was defined as circular DNAs overlapping with copy number-amplified regions identified in bulk sequencing (green) and mtDNA or chrM (red). 'Others' are defined as all other small circular DNAs (blue). Data are presented as the mean ± s.e.m.

## Intercellular differences in ecDNA content drive expression differences

The unequal mitotic segregation of ecDNA implies that ecDNA copy number can vary greatly between single cells[17,22]. In most single cells, multifragment ecDNAs did not differ in structure and composition (Fig. 3a,b), suggesting that ecDNA is structurally stable in cultured cell lines. As predicted by their binomial mitotic segregation and the conferred strong fitness advantage[2,17], most single TR14 cells contained all three independent oncogene-harboring ecDNAs also detected in bulk populations (Fig. 3b and Fig. 4a). However, a small number of cells only contained a subset of independent ecDNAs (Fig. 4a–c). This suggests that ecDNA content variation serves as a source of population heterogeneity. Intriguingly, *MDM2*-harboring ecDNAs were detected in all single cells, whereas *CDK4*- and *MYCN*-harboring ecDNAs were absent in some cells (Fig. 4b,c), suggesting that yet undefined biological principles of ecDNA segregation may exist. Next, we asked whether ecDNA copy number heterogeneity influenced the expression of genes encoded on ecDNA. We confirmed that the distribution of relative ecDNA copy number was consistent with copy number distributions measured using FISH (Supplementary Fig. 8a–h). Phasing of SNPs suggested that ecDNAs are of mono-allelic origin in each single cancer cell (Supplementary Fig. 9a,b), confirming previous observation in bulk cell populations[3]. Consistent with copy number-driven differences in gene expression, relative ecDNA copy number was positively correlated with the mRNA read counts of genes contained on ecDNAs in the same single cells (Fig. 4d–h). Even though enhancer interactions in clustered ecDNA may also contribute to intercellular ecDNA expression variability[23], we provide evidence that ecDNA copy number heterogeneity is a major determinant of intercellular differences in oncogene expression.

## scEC&T-seq detects single-nucleotide variants on ecDNA and mtDNA

Single-nucleotide variants (SNVs) are important drivers of intercellular heterogeneity and tumor evolution[31]. Furthermore, SNVs can be tracked in cells, allowing their use for lineage tracing applications[32]. To test whether scEC&T-seq could be used to detect SNVs, we applied SNV detection algorithms on merged single-cell scEC&T-seq data and compared the detected SNVs to those identified in the whole-genome sequences of bulk populations. Most SNVs detected using scEC&T were also detected in whole genomes (>69.5%). Because scEC&T-seq also detects mtDNA (Fig. 2c,d), we hypothesized that heteroplasmic mitochondrial mutations may enable lineage tracing, as demonstrated in other single-cell assays in the past[32] (Fig. 1c,d and Supplementary Fig. 1c). Indeed, unsupervised hierarchical clustering by homoplasmic mtDNA variants accurately genotyped cells (Supplementary Fig. 10a). Heteroplasmic SNVs on mtDNA revealed high intercellular heterogeneity, and unsupervised hierarchical clustering on individual single cells grouped them, which indicates subclonality and may allow lineage tracing (Supplementary Fig. 10b and Supplementary Fig. 11a,b). Thus, scEC&T-seq can detect heteroplasmic variants in mtDNA and ecDNA, allowing for a wide range of SNV-based applications and analyses, including lineage inference.

## Distinct pathways are active in cells with high small circular DNA content

Whereas the origin and functional consequences of large oncogene-containing ecDNA elements has been studied in some detail in the past[33,34], it is largely unclear how small circular DNAs are formed and how they influence the behavior of cells. Recent work suggests that some small circular elements are formed during apoptosis[6]. Other reports

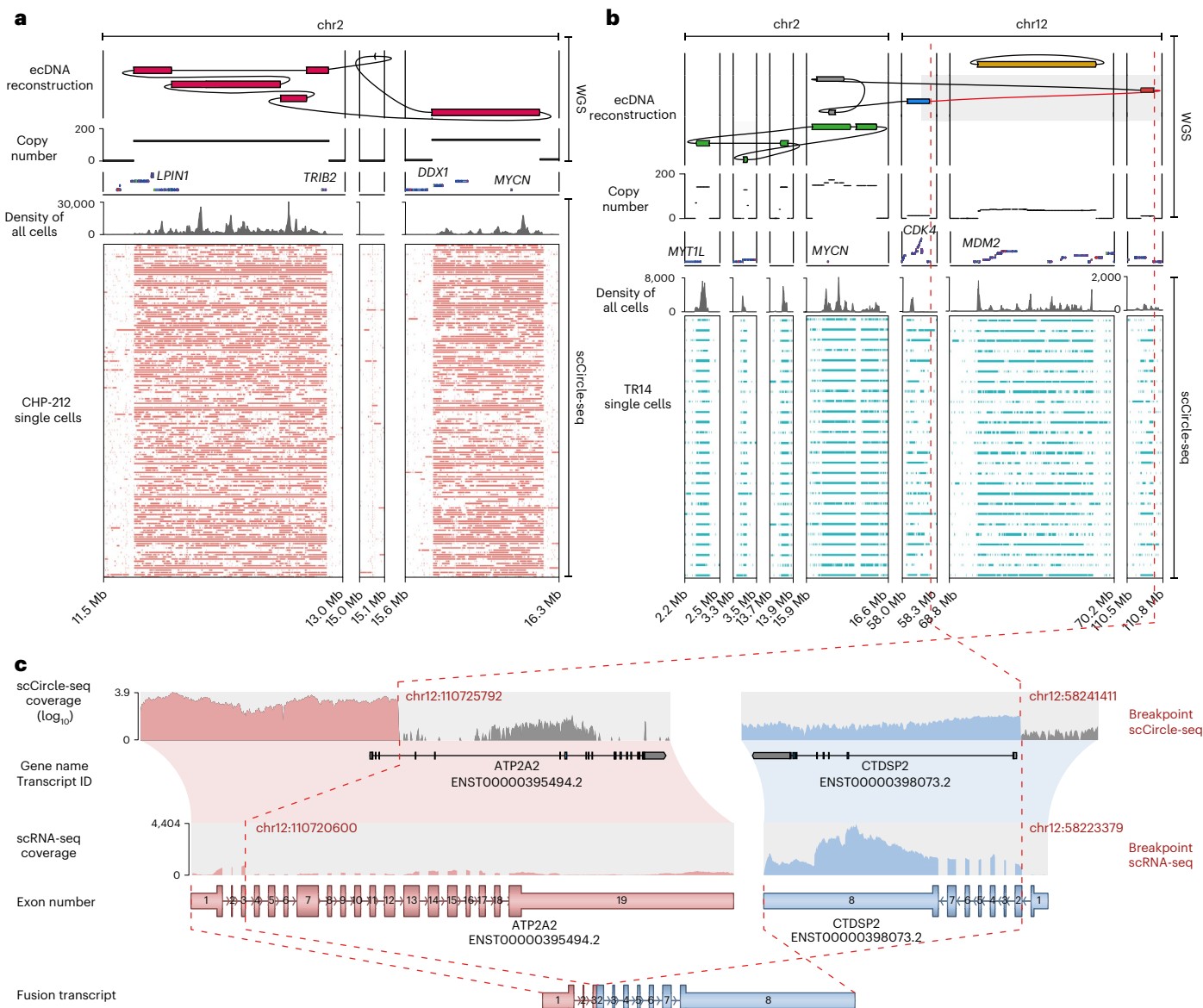

**Fig. 3 | scEC&T-seq captures the complex structure of multifragmented ecDNAs in single neuroblastoma cells. a,b,** Long- and short-read-based ecDNA reconstructions derived from WGS data in bulk cell populations and read coverage over the ecDNA fragments across single cells in CHP-212 (*n* = 150) (**a**) and TR14 (*n* = 25) cells (**b**) as detected by scEC&T-seq. Top to bottom, ecDNA amplicon reconstruction, copy number profile, gene annotations, read density over the ecDNA region in merged single cells and coverage over the ecDNA region in single cells (rows). **c,** Exemplary fusion transcript detected by scEC&T-seq

resulting from the rearrangement of chromosomal segments in the *CDK4* ecDNA in TR14. Top to bottom, scCircle-seq read coverage over the breakpoint region in merged TR14 single cells (log-scaled), transcript annotations, scRNA-seq read coverage over the fused transcripts in merged TR14 single cells, native transcript representations and fusion transcript representation. The interconnected genomic segments in *CDK4* ecDNA that give rise to the fusion gene are indicated by a red dashed line.

provide evidence for the involvement of aberrant DNA damage repair in their generation[35]. In line with previous reports[36], we identified the presence of microhomology at circular breakpoints of small circular DNAs, suggesting that microhomology-mediated repair may be involved in their generation (Supplementary Fig. 12). The bimodal size distribution identified in single cells (Fig. 2a) suggested that at least two types of small circular DNAs exist in cells. Very small circular DNAs (<3 kb) were found in all analyzed single cells (Fig. 2a and Fig. 5a). No difference was observed in the fraction of very small circular DNAs between cells at different cell cycle phases (Fig. 5b), raising the question whether such small circular DNAs can be replicated. To identify the pathways associated with the high contents of these very small circular DNAs, we compared RNA expression of cells with a high relative amount of such small circular DNAs to that of

cells with low relative content (Fig. 5a). Twenty pathways were significantly positively enriched in cell transcriptomes with high very small circular DNA content (Fig. 5c–e and Supplementary Table 6). In agreement with previous studies, DNA damage and repair pathways[35,37,38], apoptosis[6] and telomere maintenance[39] were significantly enriched in cells with a high relative content of this smaller subtype of circular DNA (Fig. 5c–e). This demonstrates that scEC&T-seq can help address long-standing questions about the origin and functional consequences of small circular DNAs.

## Small circular DNA breakpoints frequently overlap with CCCTC-binding factor sites

Chromatin conformation and accessibility can influence DNA damage susceptibility[40]. We hypothesized that small circular DNAs

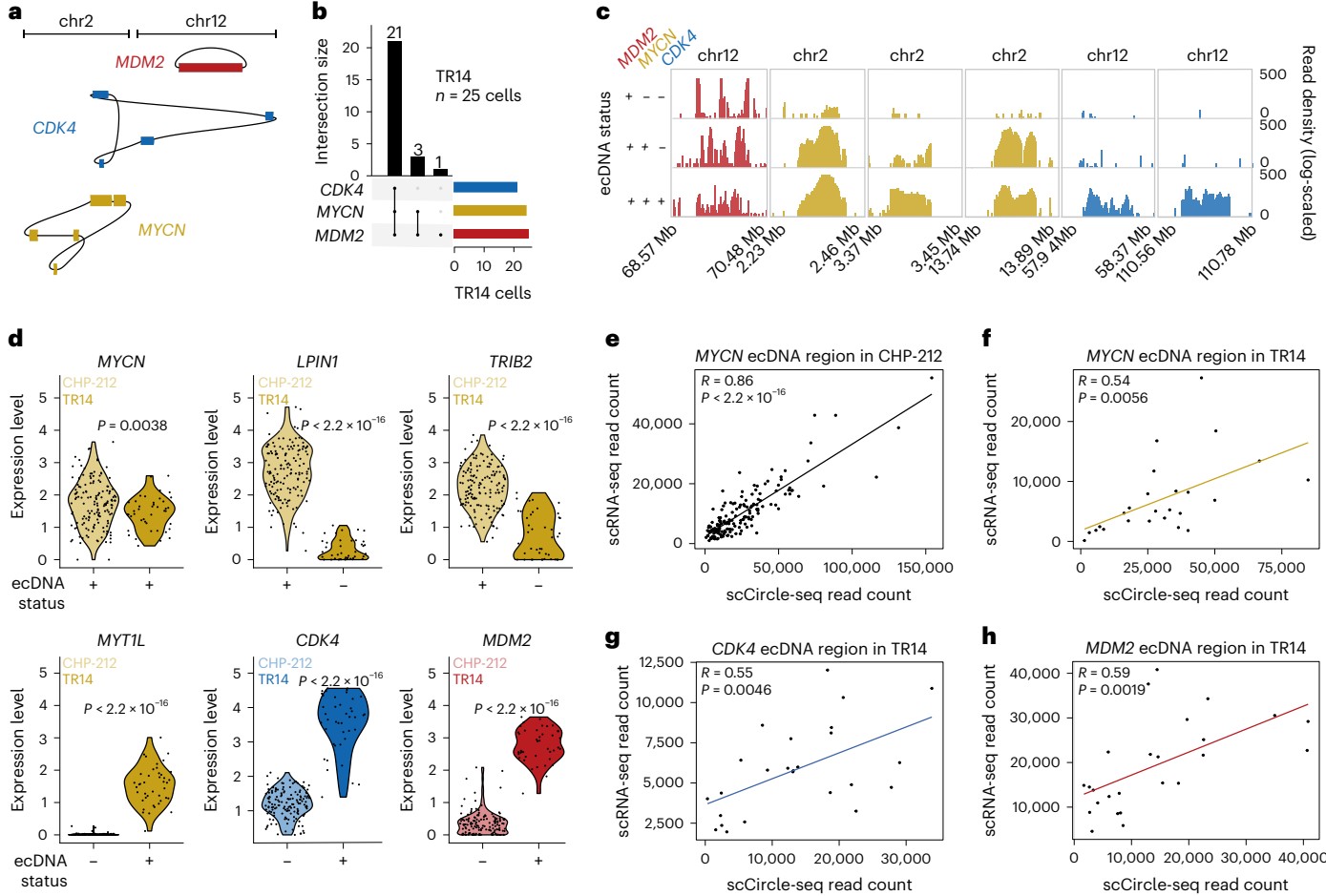

**Fig. 4 | Intercellular differences in ecDNA content drive gene expression differences. a**, Schematic representation of the three independent ecDNAs identified in TR14: *MYCN* ecDNA (yellow); *CDK4* ecDNA (blue); and *MDM2* ecDNA (red). **b**, UpSet plot displaying the co-occurrence of the three ecDNAs identified in TR14 (*MDM2, CDK4, MYCN*) in single cells (*n* = 25 TR14 cells). **c**, Genome tracks with read densities (log-scaled) over reconstructed ecDNA regions in three exemplary TR14 cells showing different ecDNAs detected. **d**, Violin plots of mRNA expression levels in TR14 and CHP-212 single cells (two-sided Welch's *t*-test; $P = 0.0038$ (*MYCN*), $P < 2.2 \times 10^{-16}$ (*LPIN1, TRIB2, CDK4, MDM2, MYT1L*));

$n = 171$ CHP-212 cells, $n = 42$ TR14 cells. **e**, **f**, Pairwise comparison between ecDNA and mRNA read counts from scEC&T-seq over the reconstructed *MYCN* ecDNA region in CHP-212 single cells (two-sided Pearson correlation, $P < 2.2 \times 10^{-16}$, $R = 0.86$, $n = 150$ cells) (**e**) and in TR14 single cells (two-sided Pearson correlation, $P = 0.0056$, $R = 0.54$, $n = 25$ cells) (**f**). **g**, **h**, Pairwise comparison between ecDNA and mRNA read counts from scEC&T-seq over the reconstructed *CDK4* (**g**) and *MDM2* (**h**) ecDNAs in TR14 single cells (two-sided Pearson correlation, $P = 0.0046$, $R = 0.55$ for *CDK4* and $P = 0.0019$, $R = 0.59$ for *MDM2*, $n = 25$ TR14 cells).

may be a product of DNA damage at sites of differential chromatin accessibility or conformation. To test this hypothesis, we measured the relative enrichment of CCCTC-binding factor (CTCF) chromatin immunoprecipitation followed by sequencing (ChIP–seq) and assay for transposase-accessible chromatin using sequencing (ATAC–seq) peaks in regions of small circular DNAs compared to other sites in the genome, respectively. Small circular DNAs detected using scEC&T-seq in single CHP-212 cells and those detected using Circle-seq in the bulk cell populations were used for this analysis (Supplementary Fig. 13a–d). Intriguingly, circular DNA breakpoints were significantly enriched at CTCF binding sites both in single cells and in bulk cell populations. This enrichment was even more striking considering that regions from which small circular DNAs originated were significantly depleted at sites of high ATAC–seq signals (Supplementary Fig. 13e). This suggests that CTCF binding sites and non-accessible chromatin, which is abundant at CTCF bindings sites[41], may be susceptible to breakage and circular DNA formation. To control for background ChIP–seq signals, we measured the enrichment of H3K4me1, H3K27ac and H3K27me3 ChIP–seq peaks at sites of small circular DNA formation. In all cases, small circular DNAs were found at significantly lower frequency at these sites than expected for randomly distributed regions (Supplementary Fig. 13f–h),

confirming the specificity of CTCF enrichment and indicating that sites marked by H3K4me1, H3K27ac and H3K27me3 may be protected from breakage and circularization. Considering the role of CTCF in regulating the three-dimensional structure of chromatin through mediation of chromatin loop formation[41], our data raise the possibility that DNA breaks during CTCF-mediated loop extrusion may represent a mechanism of small circular DNA formation.

**scEC&T-seq profiles circular DNA in primary neuroblastomas**

We next applied scEC&T-seq to single nuclei from two neuroblastomas and live T cells isolated from the blood samples of two patients (Fig. 6a, Supplementary Figs. 14a,b and 15a–t and Supplementary Note 1). The number of individual circular DNA elements identified in cancer cells was significantly higher compared to that of normal T cells and cell line cells, suggesting that DNA circularization is more frequent in tumors than in untransformed cells or cells in culture (Fig. 6b). Circular DNA size distributions and relative genomic content were comparable to those observed in cell lines, suggesting that scEC&T-seq reproducibly captures circular DNA regardless of the input material (Fig. 6b and Supplementary Figs. 4a and 16a). In agreement with our observations in cell lines, the proportion of recurrently identified small circular DNAs was

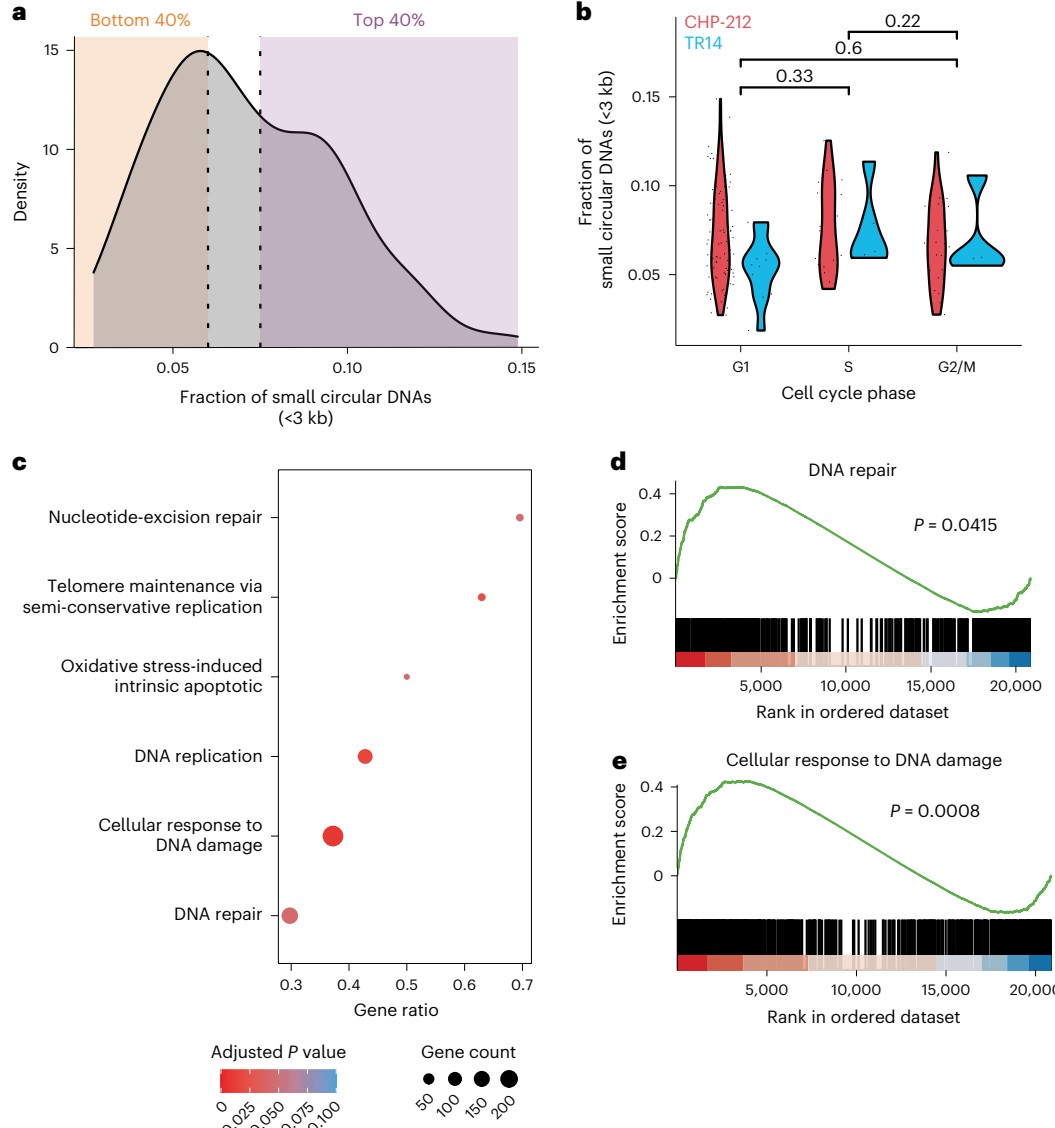

**Fig. 5 | High relative content of small circular DNAs is associated with DNA damage response pathway activation. a**, Density plot of relative small circular DNA (<3 kb) content in CHP-212 single cells ($n = 129$). For differential expression analyses, cells were divided in two categories: 'low' (orange area, bottom 40%) and 'high' (purple area, top 40%). **b**, Violin plot comparing the relative number of small circular DNAs (<3 kb) at different cell cycle phases in CHP-212 (red, $n = 129$) and TR14 (blue, $n = 20$) single cells. A two-sided Welch's $t$-test was used among the indicated conditions. $P$ values are shown. **c**, Cellular processes significantly enriched in CHP-212 cells with high relative very small circular DNA content. Adjusted $P$ values and gene counts are shown. **d**, Gene set enrichment analysis (GSEA) plot of genes involved in DNA repair (adjusted $P = 0.0415$). **e**, GSEA plot of genes involved in the cellular response to the DNA damage stimulus (adjusted $P = 0.0008$). $P$ values were adjusted using the Bejamini–Hochberg method.

low (Supplementary Fig. 16b–d). Large, oncogene-containing ecDNAs, on the other hand, were recurrently identified in tumor nuclei but not in T cells (Fig. 6c and Supplementary Fig. 16b–d), in agreement with their oncogenic role. *MYCN*-containing ecDNAs were detectable in almost all cancer nuclei from both patients, which was confirmed with FISH (Supplementary Fig. 16e–g). As observed in cell lines, intercellular differences in *MYCN* transcription positively correlated with relative ecDNA content (Supplementary Fig. 16h,i). Thus, scEC&T-seq can be successfully applied to human tumors.

**scEC&T-seq enables inference of ecDNA structural dynamics**
Recent studies of cancer genomes have described structurally complex ecDNAs[3,11,18,19,28,29,42]; however, due to the analysis of bulk cell populations, they were limited in their ability to infer structural ecDNA heterogeneity. Both analyzed neuroblastomas contained large and

structurally complex *MYCN*-containing ecDNAs, as confirmed using long-read Nanopore sequencing of the same single nuclei and by whole-genome sequencing (WGS) of bulk cell populations (Fig. 7a and Supplementary Fig. 17a). Whereas the ecDNA structure in patient no. 1 was so complex that it was not fully computationally reconstructed (Supplementary Fig. 17b), the *MYCN*-containing ecDNA in the other patient (patient no. 2) was structurally composed of five individual genomic fragments, all derived from chromosome 2, which were connected by four SVs (nos. 1–4) in a manner that was simple enough to be reliably reconstructed in single cells (Fig. 7a). We hypothesized that the assessment of intercellular ecDNA structural heterogeneity in this patient could facilitate the inference of ecDNA structural dynamics. Indeed, ecDNA considerably structurally differed between a subset of single cells (Fig. 7a,b). SV no. 1 was present in all single cells, suggesting it occurred before the other SVs and may represent the initial variant

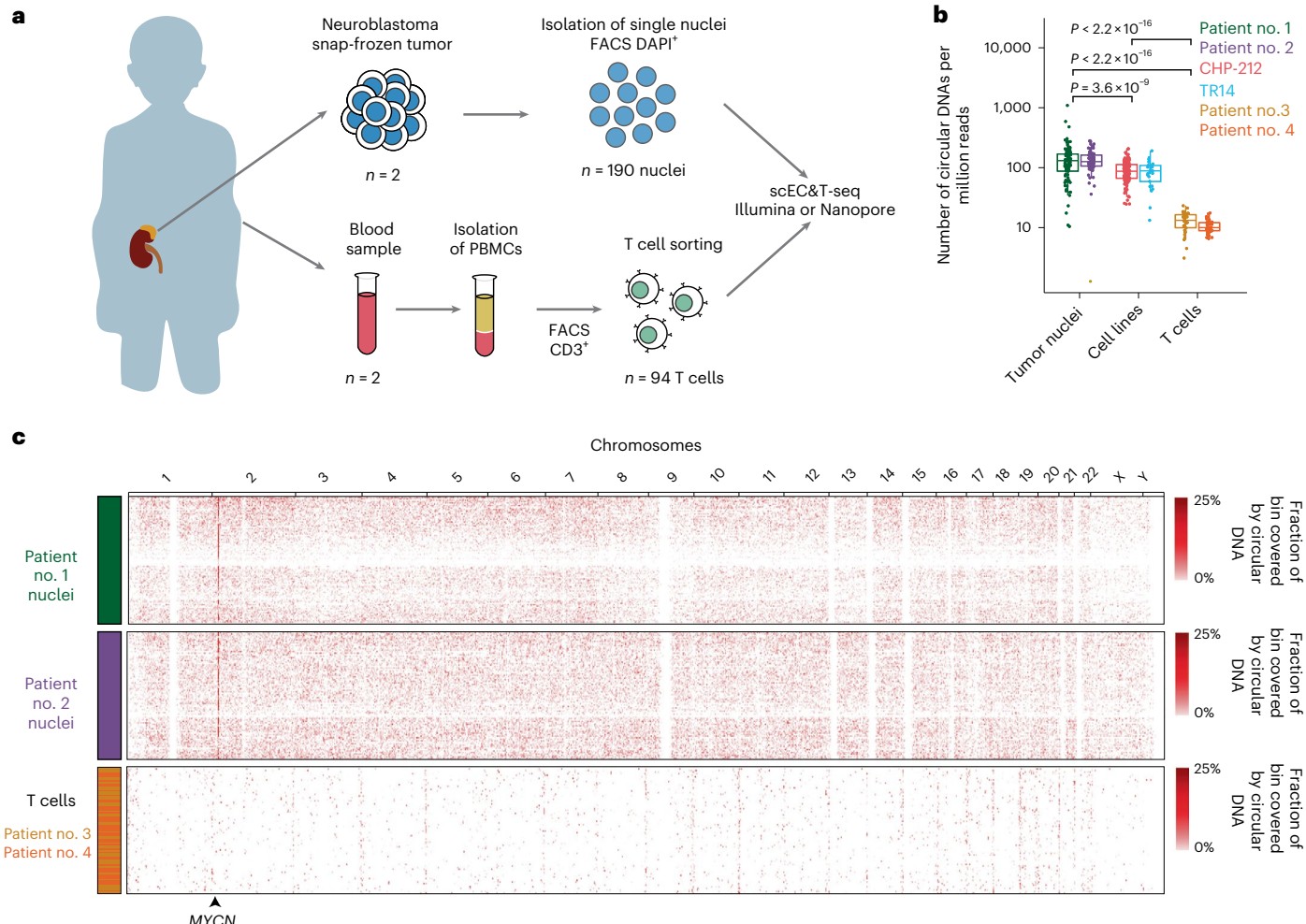

**Fig. 6 | scEC&T-seq detects circular DNAs in primary neuroblastomas at the single-cell level. a**, Schematic diagram describing tumor and blood sample processing. **b**, Number of individual circular DNA regions normalized by library size detected in primary tumor nuclei (*n* = 93 nuclei patient no. 1, *n* = 86 nuclei patient no. 2), neuroblastoma cell line single cells (*n* = 25 TR14 cells, *n* = 150 CHP-212 cells) and nonmalignant single T cells (*n* = 38 patient no. 3, *n* = 41 patient no. 4). *P* values were calculated using a two-sided Welch's *t*-test and are shown.

The boxes in the boxplots represent the 25th and 75th percentiles with the center bar as the median value and the whiskers representing the furthest outlier ≤1.5× the IQR from the box. **c**, Heatmap of the genome-wide circular DNA density in neuroblastoma primary tumors and normal T cells (*n* = 93 patient no. 1, green; *n* = 86 patient no. 2, purple; *n* = 38 patient no. 3, yellow; *n* = 41 patient no. 4, orange; bin sizes = 3 Mb). The location of the *MYCN* gene in chr2 is shown.

leading to circularization (Fig. 7b–d). SVs nos. 2–4, on the other hand, were not detected in a subset of cells. Moreover, SV no. 2 and SV no. 3 indicated the presence of a 6-kb deletion and SV no. 4 supported the presence of a larger deletion (approximately 180 kb) on the ecDNA, both of which were present in most but not all single cells (94.2%; Fig. 7c,d). Analysis of split reads at the breakpoints of SV nos. 2 and 3, that is, the edges of the 6-kb deletion, and coverage across this deletion in single cells, suggested the presence of three different subclonal cell populations we termed subclone nos. 1–3. Clone no. 1 contained an intact ecDNA lacking deletions. Clone no. 2 harbored a mixed population of ecDNAs with and without deletions (Fig. 7b–e). In clone no. 3, the detected SVs and sequencing coverage indicated the presence of a pure population of ecDNAs harboring both deletions and all SVs (Fig. 7c–e). The simplest sequence of mutational events that would result in the observed intercellular structural ecDNA heterogeneity starts with a simple excision of an ecDNA containing *MYCN* and neighboring chromosomal regions, that is, SV no. 1 generating ecDNA variant no. 1 found in clone no. 1 (Fig. 7e,f). This is followed by the fusion of two simple ecDNA no. 1 variants generating a more complex rearranged ecDNA variant no. 2 that includes the small 6-kb deletion and SV nos. 2

and 3 in addition to SV no. 1 (Fig. 7e,f). Such circular recombination is in agreement with recent models based on WGS[43]. An additional large deletion on this ecDNA would create ecDNA variant 3 with all SV nos. 1–4 and both deletions (Fig. 7e,f). The predominance of ecDNA variant 3 in these neuroblastoma cells suggests that it may confer a positive selective advantage. Our proof-of-principle demonstration that scEC&T-seq can help infer ecDNA structural dynamics illustrates that scEC&T-seq may facilitate future studies addressing important open questions about the origin and evolution of ecDNA.

## Enhancers are coamplified with oncogenes on ecDNA in single cells

Regulatory elements are commonly amplified on ecDNA, have an essential role in the transcriptional regulation of oncogenes on ecDNA and are assumed to be under strong positive selection[28,29]. Indeed, at least one of the recently described *MYCN*-specific enhancer elements[28,29] was recurrently detected on ecDNAs harboring *MYCN* in over 82.7% of neuroblastoma single cells (Fig. 7f and Supplementary Fig. 18a). Interestingly, the deletion detected in patient no. 2, that is, ecDNA variant 3, is predicted to result in the loss of one of two *MYCN*

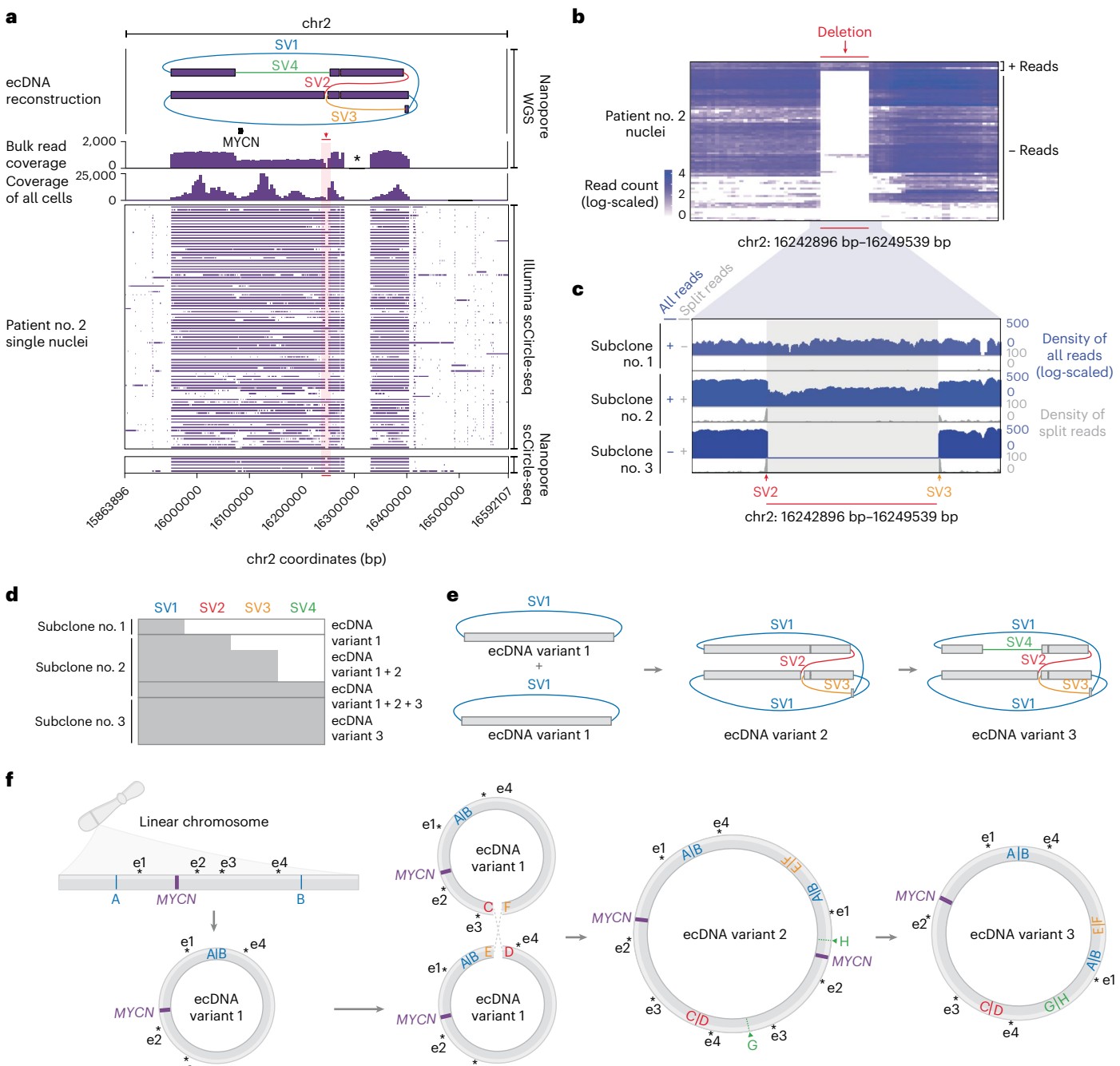

**Fig. 7 | scEC&T-seq profiles intercellular structural ecDNA heterogeneity in neuroblastomas. a**, Long read-based ecDNA reconstructions derived from WGS data in bulk populations and read coverage over the ecDNA fragments across single nuclei in patient no. 2 (*n* = 86 nuclei) as detected by long-read or short-read scEC&T-seq. Top to bottom, ecDNA amplicon reconstruction (the SVs on ecDNAs are colored; SV nos. 1–4), gene annotation, read density over the ecDNA region in bulk long-read Nanopore WGS data, read density over the ecDNA region in merged single nuclei and coverage over the ecDNA region in single nuclei (rows) as detected by long-read or short-read scEC&T-seq. The 6-kb deletion is highlighted in red. The single asterisk indicates the unmappable region of the reference genome (hg19). **b**, Heatmap of the total number of reads (log-scaled) in a 500-bp window over the identified 6-kb deletion on ecDNA across single nuclei

in patient no. 2 (*n* = 86 nuclei). **c**, Exemplary genome tracks of the three identified clone variants in patient no. 2 based on the absence or presence of the 6-kb deletion on the ecDNA element. The log-scaled total read density is shown in blue and the circle edge-supporting read density is shown in gray. **d**, Detection of SV nos. 1–4 supporting the multifragmented ecDNA element in eight exemplary single cells representing the three identified clone variant groups (≥1 read supporting the SV, gray; 0 reads supporting the SV, white). **e**, Schematic representation of ecDNA variants 1–3 detected in **d**. **f**, Schematic interpretation of the evolution of the ecDNA structure in patient no. 2 based on the identified ecDNA variants in the scEC&T-seq data. The position of the *MYCN* oncogene and its local enhancer elements (e1–e5), indicated by the single asterisks, in each ecDNA variant is shown.

gene copies, including regulatory elements e2 and e3 present on ecDNA variant 2 (Fig. 7f). This raises the possibility that the change in enhancer:oncogene stoichiometry (6:1 in variant 3 versus 8:2 in variant 2), that is, the presence of one instead of two oncogene copies

on an ecDNA, may be beneficial for oncogene expression because it may allow a more efficient use of enhancers on the ecDNA. Such mechanisms may explain the observed predominance of ecDNA variant no. 3 in the tumor cell population.

Recent reports suggest that ecDNAs not harboring oncogenes but containing enhancer elements exist and can enhance transcriptional output on linear chromosomes or on other ecDNAs in *trans* as part of ecDNA hubs[17,23]. To identify such ecDNA elements, we analyzed H3K4me1, H3K27ac, H3K27me3 ChIP–seq and ATAC–seq data from neuroblastoma cells and searched for ecDNAs including these regions but not harboring oncogenes. No ecDNA only harboring enhancer elements was recurrently identified in single neuroblastoma cells. All recurrently detected ecDNAs contained at least one oncogene. However, a large set of nonrecurrent small circular DNAs were identified that only contained genomic regions with regulatory elements (Supplementary Fig. 18b). The lack of recurrence of these circular DNA elements, however, suggests that they are not maintained in these cancer cells or do not confer positive selective advantages. Thus, scEC&T-seq allows the detection of noncoding circular DNAs and enables future investigations of their role in transcriptional regulation in cancer.

## Discussion

We have shown that by parallel sequencing of circular DNA and mRNA from single cancer cells, scEC&T-seq not only readily distinguishes the transcriptional consequences of ecDNA-driven intercellular oncogene copy number heterogeneity, but also has the potential to uncover principles of ecDNA structural evolution. We believe that the integrated analysis of a cell's circular DNA content and transcriptome through scEC&T-seq will enable a more complete understanding of the extent, function, heterogeneity and evolution of circular DNAs in cancer and beyond.

scEC&T-seq complements recently published methods for single-cell DNA and single-cell RNA sequencing (scRNA-seq)[23,27], which cannot readily distinguish linear intra- from extrachromosomal circular amplicons. Even though scEC&T-seq is compatible with automation, the elaborate circular DNA enrichment procedures only allow low throughput, which drives costs per cell and currently represents a limitation of this method. However, compared to droplet-based microfluidic single-cell technologies, plate-based scEC&T-seq generates a uniform number of reads per cell and produces independent sequencing libraries available for selection and resequencing, which is advantageous when high sequencing coverage is needed. Indeed, we showed that scEC&T can be combined with different sequencing technologies. The level of detail provided by scEC&T-seq far exceeds that of high-throughput methods. Pairing our method with other single-cell technologies, for example, Strand-seq[44], and processing approaches, for example, single-cell tri-channel processing[45], may increase the spectrum of somatic variation detected by scEC&T-seq.

Performing scEC&T-seq in single cancer cells allowed us to profile their circular DNA content independently of copy number and circular DNA size. Small circular DNAs were identified in live single cells, suggesting that apoptosis is not the only mechanism of their generation. Whereas oncogene-containing ecDNAs were clonally present in single cells, small circular DNAs were exclusive to single cells. This not only indicates that small circular DNAs probably do not confer a selective advantage to cancer cells, but also suggests the existence of yet unknown prerequisites for selection, propagation and maintenance of these circular DNAs.

The robust demonstration of integrating circular DNA and mRNA sequencing in single cancer cells indicates that the same approach can be applied to a diverse range of biological systems to further explore the diversity and invariance of circular DNA in single cells. Thus, we anticipate that our method will be a resource for future research in many fields beyond cancer biology and suggest that it has the potential to address many currently unresolved biological questions regarding circular DNA.

## Online content

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

[1]Department of Pediatric Oncology and Hematology, Charité – Universitätsmedizin Berlin, corporate member of Freie Universität Berlin, Humboldt-Universität zu Berlin, Berlin, Germany. [2]Experimental and Clinical Research Center of the MDC and Charité Berlin, Berlin, Germany. [3]Genomics Technology Platform, Max Delbrück Center for Molecular Medicine in the Helmholtz Association, Berlin, Germany. [4]Berlin Institute for Medical Systems Biology, Max Delbrück Center for Molecular Medicine in the Helmholtz Association, Berlin, Germany. [5]Charité–Universitätsmedizin Berlin, Berlin, Germany. [6]Faculty of Life Science, Humboldt-Universität zu Berlin, Berlin, Germany. [7]Freie Universität Berlin, Berlin, Germany. [8]Fraunhofer Institute for Cell Therapy and Immunology, Branch Bioanalytics and Bioprocesses IZI-BB, Potsdam, Germany. [9]Max-Delbrück-Centrum für Molekulare Medizin, Berlin, Germany. [10]RG Development and Disease, Max Planck Institute for Molecular Genetics, Berlin, Germany. [11]Institute for Medical Genetics, Charité–Universitätsmedizin Berlin, Berlin, Germany. [12]Center for Personal Dynamic Regulomes, Stanford University School of Medicine, Stanford, CA, USA. [13]Berlin-Brandenburg Center for Regenerative Therapies, Charité–Universitätsmedizin Berlin, Berlin, Germany. [14]Howard Hughes Medical Institute, Stanford University School of Medicine, Stanford, CA, USA. [15]German Cancer Consortium, partner site Berlin, and German Cancer Research Center, Heidelberg, Germany. [16]Berlin Institute of Health, Berlin, Germany. [17]Institute for Computational Cancer Biology, Center for Integrated Oncology, Cancer Research Center Cologne Essen Faculty of Medicine and University Hospital Cologne, University of Cologne, Cologne, Germany. [18]Berlin Institute for the Foundations of Learning and Data, Berlin, Germany. [19]Center for Epigenetics Research, Memorial Sloan Kettering Cancer Center, New York, NY, USA. [20]These authors jointly supervised this work: Richard P. Koche and Anton G. Henssen. ✉e-mail: henssenlab@gmail.com

## Methods

### scEC&T sequencing

A detailed, step-by-step protocol of scEC&T-seq is available on the Nature Protocol Exchange[46] and is described below. The duration of the protocol is approximately 8 days per 96-well plate.

### Cell culture

Human tumor cell lines were obtained from ATCC (CHP-212) or were provided by J. J. Molenaar (TR14; Princess Máxima Center for Pediatric Oncology). The identity of all cell lines was verified by short tandem repeat genotyping (Genetica DNA Laboratories and IDEXX BioResearch); absence of *Mycoplasma* spp. contamination was determined with a Lonza MycoAlert Detection System. Cell lines were cultured in Roswell Park Memorial Institute 1640 medium (Thermo Fisher Scientific) supplemented with 1% penicillin, streptomycin and 10% FCS. To assess the number of viable cells, cells were trypsinized (Gibco), resuspended in medium and sedimented at 500 g for 5 min. Cells were then resuspended in medium, mixed in a 1:1 ratio with 0.02% trypan blue (Thermo Fisher Scientific) and counted with a TC20 cell counter (Bio-Rad Laboratories).

### Preparation of metaphase spreads

Cells were grown to 80% confluency in a 15-cm dish and metaphase-arrested by adding KaryoMAX Colcemid (10 µl ml⁻¹, Gibco) for 1–2 h. Cells were washed with PBS, trypsinized (Gibco) and centrifuged at 200g for 10 min. We added 10 ml of 0.075 M KCl preheated at 37 °C, 1 ml at a time, vortexing at maximum speed in between. Afterwards, cells were incubated for 20 min at 37 °C. Then, 5 ml of ice-cold 3:1 MeOH:acetic acid (kept at −20 °C) were added, 1 ml at a time followed by resuspension of the cells by flicking the tube. The sample was centrifuged at 200g for 5 min. Addition of the fixative followed by centrifugation was repeated four times. Two drops of cells within 200 µl of MeOH:acetic acid were dropped onto prewarmed slides from a height of 15 cm. Slides were incubated overnight.

### FISH

Slides were fixed in MeOH:acetic acid for 10 min at −20 °C followed by a wash of the slide in PBS for 5 min at room temperature. Slides were incubated in pepsin solution (0.001 N HCl) with the addition of 10 µl pepsin (1 g 50 ml⁻¹) at 37 °C for 10 min. Slides were washed in 0.5× saline-sodium citrate (SSC) buffer for 5 min and dehydrated by washing in 70%, 90% and 100% cold ethanol (stored at −20 °C) for 3 min. Dried slides were stained with 10 µl Vysis LSI N-MYC SpectrumGreen/CEP 2 SpectrumOrange Probes (Abbott), ZytoLight SPEC CDK4/CEN 12 Dual Color Probe (ZytoVision) or ZytoLight SPEC MDM2/CEN 12 Dual Color Probe (ZytoVision), covered with a coverslip and sealed with rubber cement. Denaturing occurred in a ThermoBrite system (Abbott) for 5 min at 72 °C followed by 37 °C overnight incubation. The slides were washed for 5 min at room temperature in 2× SSC/0.1% IGEPAL, followed by 3 min at 60 °C in 0.4× SSC/0.3% IGEPAL (Sigma-Aldrich) and an additional wash in 2× SSC/0.1% IGEPAL for 3 min at room temperature. Dried slides were stained with 12 µl Hoechst 33342 (10 µM, Thermo Fisher Scientific) for 10 min and washed with PBS for 5 min. After drying, a coverslip was mounted on the slide and sealed with nail polish. Images were taken using a Leica SP5 Confocal microscope (Leica Microsystems).

### Interphase FISH

CHP-212 and TR14 cells for the interphase FISH were grown in 8-chamber slides (Nunc Lab-Tek, Thermo Scientific Scientific) to 80% confluence. Wells were fixed in MeOH:acetic acid for 20 min at −20 °C followed by a PBS wash for 5 min at room temperature. The wells were removed and the slides were digested in pepsin solution (0.001 N HCl) with the addition of 10 µl pepsin (1 g 50 ml⁻¹) at 37 °C for 10 min. After a wash in 0.5× SSC for 5 min, slides were dehydrated by washing in 70%, 90% and 100% cold ethanol stored at −20 °C (3 min in each solution). Dried slides were stained with either 5 µl Vysis LSI N-MYC SpectrumGreen/CEP 2 SpectrumOrange Probes, ZytoLight SPEC CDK4/CEN 12 Dual Color Probe or ZytoLight SPEC MDM2/CEN 12 Dual Color Probe, covered with a coverslip and sealed with rubber cement. Denaturing occurred in a ThermoBrite system for 5 min at 72 °C followed by 37 °C overnight. Slides were washed for 5 min at room temperature within 2× SSC/0.1% IGEPAL, followed by 3 min at 60 °C in 0.4× SSC/0.3% IGEPAL and a further 3 min in 2× SSC/0.1% IGEPAL at room temperature. Dried slides were stained with 12 µl Hoechst 33342 (10 µM) for 10 min and washed with PBS for 5 min. After drying, a coverslip was mounted on the slide and sealed with nail polish. Images were taken with a Leica SP5 Confocal microscope. For ecDNA copy number estimation, we counted foci using FIJI v.2.1.0 with the function find maxima. Nuclear boundaries were marked as regions of interest. The threshold for signal detection within the regions of interest was determined manually and used for all images analyzed within one group.

### Patient samples and clinical data access

This study includes tumor and blood samples of patients diagnosed with neuroblastoma between 1991 and 2022. Patients were registered and treated according to the trial protocols of the German Society of Pediatric Oncology and Hematology (GPOH). This study was conducted in accordance with the World Medical Association Declaration of Helsinki (2013 version) and good clinical practice; informed consent was obtained from all patients or their guardians. The collection and use of patient specimens was approved by the institutional review boards of Charité-Universitätsmedizin Berlin and the Medical Faculty at the University of Cologne. Specimens and clinical data were archived and made available by Charité-Universitätsmedizin Berlin or the National Neuroblastoma Biobank and Neuroblastoma Trial Registry (University Children's Hospital Cologne) of the GPOH. The *MYCN* copy number was determined using FISH. Tumor samples presented at least 60% tumor cell content as evaluated by a pathologist.

### Isolation of nuclei

Tissue samples were homogenized using a precooled glass dounce tissue homogenizer (catalog no. 357538, Wheaton) in 1 ml of ice-cold EZ PREP buffer (Sigma-Aldrich). Ten strokes with a loose pestle followed by five additional strokes with a tight pestle were used for tissue homogenization. To reduce the heat caused by friction, the douncer was always kept on ice during homogenization. The homogenate was filtered through a Falcon tube (Becton Dickinson) with a 35-µm cell strainer cap. The number of intact nuclei was estimated by staining and counting with 0.02% trypan blue (Thermo Fisher Scientific) mixed in a 1:1 ratio.

### Isolation of peripheral blood mononuclear cells from blood samples

Peripheral blood mononuclear cells (PBMCs) were isolated using density gradient centrifugation with Ficoll-Plaque PLUS (Cytiva). Whole-blood samples were resuspended 1:1 in calcium-free PBS and slowly added to 12 ml of Ficoll-Plaque PLUS. The sample was centrifuged at 200g for 30 min without breaking. The upper layer of PBMCs was isolated and washed into 40 ml of PBS. PBMCs were collected by centrifugation at 500g for 5 min and resuspended in 10% dimethylsulfoxide in FCS. The PBMC suspensions were stored at −80 °C until use.

### FACS

For single-cell sorting, 1–10 million neuroblastoma cells or PBMCs were stained with propidium iodide (PI) (Thermo Fisher Scientific) in 1× PBS; viable cells were selected based on forward and side scattering properties and PI staining. PBMC suspensions were additionally stained with a 1:400 dilution of anti-human CD3 (Ax700, BioLegend). Nuclei suspensions were stained with DAPI (final concentration 2 µM, Thermo

Fisher Scientific). Viable cells, CD3+ PBMCS or DAPI+ nuclei were sorted using a FACSAria Fusion Flow Cytometer (BD Biosciences) into 2.5 µl of RLT Plus buffer (QIAGEN) in low-binding 96-well plates (4titude) sealed with foil (4titude) and stored at −80 °C until processing.

### Genomic DNA and mRNA separation from single cells

Physical separation of genomic DNA (gDNA) and mRNA was performed as described previously in the G&T-seq protocol by Macaulay et al.[27]. All samples were processed using a Biomek FXP Laboratory Automation Workstation (Beckman Coulter). Briefly, polyadenylated mRNA was captured using a modified Oligo dT primer (Supplementary Table 7) conjugated to streptavidin-coupled magnetic beads (Dynabeads MyOne Streptavidin C1, catalog no. 65001, Invitrogen). The conjugated beads were directly added (10 µl) to the cell lysate and incubated for 20 min at room temperature with mixing at 800 rpm (MixMate, Eppendorf). Using a magnet (Alpaqua), the captured mRNA was separated from the supernatant containing the gDNA. The supernatant containing gDNA was transferred to a new 96-well plate (4titude); the mRNA-captured beads were washed three times at room temperature in 200 µl of 50 mM Tris-HCl (pH 8.3), 75 mM KCl, 3 mM MgCl2, 10 mM dithiothreitol (DTT), 0.05% Tween 20 and 0.2× RNase inhibitor (SUPERase•In, Thermo Fisher Scientific). For each washing step, the beads were mixed for 5 min at 2,000 rpm in a MixTape (Eppendorf). The supernatant was collected after each wash and pooled with the original supernatant using the same tips to minimize DNA loss.

### Complementary DNA generation

The mRNA captured on the beads was eluted into 10 µl of a reverse-transcription master mix including 10 U µl$^{-1}$ SuperScript II Reverse Transcriptase (Thermo Fisher Scientific), 1 U µl$^{-1}$ RNase inhibitor, 1× Superscript II First-Strand Buffer (Thermo Fisher Scientific), 2.5 mM DTT (Thermo Fisher Scientific), 1 M betaine (Sigma-Aldrich), 6 mM MgCl2 (Thermo Fisher Scientific), 1 µM template-switching oligo (Supplementary Table 7), deoxynucleoside triphosphate mix (1 mM each of dATP, dCTP, dGTP and dTTP) (Thermo Fisher Scientific) and nuclease-free water (Thermo Fisher Scientific) up to the final volume (10 µl). Reverse transcription was performed on a thermocycler for 60 min at 42 °C followed by 10 cycles of 2 min at 50 °C and 2 min at 42 °C and ending with one 10-min incubation at 60 °C. Amplification of complementary DNA (cDNA) by PCR was immediately performed after reverse transcription by adding 12 µl of PCR master mix including 1× KAPA HiFi HotStart ReadyMix with 0.1 µM ISPCR primer (10 mM; Supplementary Table 7) directly to the 10 µl of the reverse transcription reaction mixture. The reaction was performed on a thermocycler for seven cycles as follows: 98 °C for 3 min, then 18 cycles of 98 °C for 15 s, 67 °C for 20 s, 72 °C for 6 min and finally 72 °C for 5 min. The amplified cDNA was purified using a 1:0.9 volumetric ratio of Ampure Beads (Beckman Coulter) and eluted into 20 µl of elution buffer (Buffer EB, QIAGEN).

### Circular DNA isolation, amplification and purification

The isolated DNA was purified using a 1:0.8 volumetric ratio of Ampure Beads. The sample was incubated with the beads for 20 min at room temperature with mixing at 800 rpm (MixMate). Circular DNA isolation was performed as described previously in bulk populations[3,25]. Briefly, the DNA was eluted from the beads directly into an exonuclease digestion master mix (20 units of Plasmid-Safe ATP-dependent DNase (Epicentre), 1 mM ATP (Epicentre), 1× Plasmid-Safe Reaction Buffer (Epicentre)) in a 96-well plate. In a subset of samples, 1 µl of the endonuclease MssI/PmeI (20 U µl, New England Biolabs) was added. The digestion of linear DNA was performed for 1 or 5 days at 37 °C with 10 U of Plasmid-Safe DNase and 4 µl of ATP (25 mM), which was added again every 24 h to continue the enzymatic digestion. After 1 or 5 days of enzymatic digestion, the exonuclease was heat-inactivated by incubating at 70 °C for 30 min. The exonuclease-resistant DNA was purified

and amplified using the REPLIg Single-Cell Kit (QIAGEN) according to the manufacturer's instructions. For this purification step, 32 µl of poly-ethylene glycol buffer (18% (w/v) (Sigma-Aldrich), 25 M NaCl, 10 mM Tris-HCl, pH 8.0, 1 mM EDTA, 0.05% Tween 20) were added, mixed and incubated for 20 min at room temperature. After incubation, the beads were washed twice with 80% ethanol and the exonuclease-resistant DNA was eluted directly into the reaction mixture multiple displacement amplification with a REPLIg Single-Cell Kit (QIAGEN). Amplified circular DNA was purified using a 1:0.8 volumetric ratio of Ampure Beads and eluted in 100 µl of elution buffer (Buffer EB, QIAGEN).

### Library preparation and sequencing

A total of 20 ng amplified cDNA or circular DNA was used for library preparation using the NEBNext Ultra II FS (New England Biolabs) according to the manufacturer's protocol. Samples were barcoded using unique dual-index primer pairs (New England Biolabs) and libraries were pooled and sequenced on a HiSeq 4000 instrument (Illumina) or a NovaSeq 6000 instrument with 2× 150-bp paired-end reads for circular DNA libraries and 2× 75-bp paired-end reads for cDNA libraries.

### Genomic and transcriptomic read alignments

Sequenced reads from the gDNA libraries were trimmed using TrimGalore (v.0.6.4)[47] and mapped to the human genome build 19 (GRCh37/hg19). Alignment was performed with the Burrows–Wheeler Aligner (BWA)-MEM (v.0.7.17)[48]. Following the recommendation of the Human Cell Atlas project[49] (v.2.2.1)[50] was used to align the RNA-seq data obtained from Smart-seq2 (ref. [26]) against a transcriptome reference created from the hg19 and ENCODE annotation v.19 (ref. [51]). Afterwards, genes and isoforms were quantified using rsem (v.1.3.1)[52] with a single cell prior.

### Nanopore scCircle-seq

Before Nanopore sequencing, the amplified circular DNA from single cells was subjected to T7 endonuclease digestion to reduce DNA branching. Then, 1.5 µg of amplified circular DNA were incubated at 37 °C for 30 min with 1.5 µl T7 endonuclease I (10 U µl$^{-1}$, New England Biolabs) in 3 µl of NEBuffer 2 and nuclease-free water up to a final volume of 30 µl. The endonuclease-digested DNA was purified using a 1:0.7 volumetric ratio of Ampure Beads and eluted in 25 µl of nuclease-free water. Libraries were prepared using the ONT Rapid Barcoding Kit (catalog no. SQK-RBK004, Oxford Nanopore Technologies) according to the manufacturer's instructions, and sequenced on an R9.4.1 MinION flowcell (FLO-MIN106, Oxford Nanopore Technologies). A maximum of four samples were multiplexed per run.

### Nanopore scCircle-seq data processing

The scCircle-seq Nanopore data were base-called and demultiplexed using Guppy (v.5.0.14; running guppy_basecaller with dna_r9.4.1_450bps_hac model and guppy_barcoder with FLO-MIN106 and default parameters). The obtained reads were quality-filtered using NanoFilt[53] (v.2.8.0) (-l100--headcrop 50--tailcrop 50) and aligned using ngmlr[54] (v.0.2.7) against the GRCh37/hg19 reference genome. To call SVs, we applied Sniffles[54] (v.1.0.12) (--min_homo_af 0.7--min_het_af 0.1--min_length 50--min_support 4); to obtain the binned coverage, we used deepTools[55] (v.3.5.1) bamCoverage. All these steps are available as a Snakemake pipeline (https://github.com/henssen-lab/nano-wgs).

### Circle-seq in bulk populations

Circle-seq in bulk populations was performed as described previously[3]. A detailed step-by-step protocol can be found on the Nature Protocol Exchange server.

### ChIP−seq

We generated H3K27me3 ChIP−seq data for CHP-212 according to a previously described protocol[28]. Briefly, 5−10 million CHP-212 cells

were fixed in 10% FCS-PBS with 1% paraformaldehyde for 10 min at room temperature. Chromatin was prepared as described previously[28] and sheared until a fragment size of 200–500 bp. H3K27me3–DNA complexes were immunoprecipitated for 15 h at 4 °C with an anti-H3K27me3 polyclonal antibody (catalog no. 07-449, Sigma-Aldrich). In total 10–15 μg of chromatin and 2.5 μg of antibody were used for immunoprecipitation. Libraries for sequencing were prepared using Illumina Nextera adapters according to the recommendations provided. Libraries were sequenced in 50-bp single-read mode in an Illumina HiSeq 4000 sequencer. FASTQ files were quality-controlled with FASTQC (v.0.11.8) and adapters were trimmed using BBMap (v.38.58). Reads were aligned to the hg19 using the BWA-MEM[48] (v.0.7.15) with default parameters. Duplicate reads were removed using Picard (v.2.20.4).

## Chromatin marks enrichment analyses

We obtained public CHP-212 copy number variation, ChIP–seq (H3K27ac, H3K4me1, CTCF) and ATAC–seq data[28,56]. For further analysis, we used the processed bigwig tracks, filtered to exclude ENCODE Data Analysis Center (DAC) blacklisted regions and normalized to read counts per million (CPM) in 10-bp bins, and peak calls provided by Helmsauer et al.[28]. To assess the correlation of epigenetic marks with circle regions, we only considered circle regions that did not overlap with copy number variation in CHP-212 or ENCODE DAC blacklisted regions. For H3K27ac, H3K4me1 and H3K27me3 ChIP–seq and ATAC–seq data, we computed the mean CPM signal across all circle regions, weighted by the respective circle sizes. To test for statistical association, we created 1,000 datasets with randomized circle positions within a genome masked for copy number variation in CHP-212 and ENCODE DAC blacklisted regions using regioneR[57] (v.1.24.0). We derived an empirical P value from the distribution of mean CPM signal across the randomized circle regions. For CTCF ChIP–seq data, we calculated the percentage of circle edges overlapping with a CTCF peak and assessed statistical significance using the same randomization strategy as described above.

## Circle-seq analysis

Extrachromosomal circular DNA analysis was performed as described previously[3]. Reads were 3′-trimmed for both quality and adapter sequences, with reads removed if the length was less than 20 nucleotides. BWA-MEM (v.0.7.15) with default parameters was used to align the reads to the human reference assembly GRCh37/hg19; PCR and optical duplicates were removed with Picard (v.2.16.0). Putative circles were classified with a two-step procedure. First, all split reads and read pairs containing an outward-facing read orientation were placed in a new BAM file. Second, regions enriched for signal over background with a false discovery rate < 0.001 were detected in the 'all reads' BAM file using variable-width windows from Homer v.4.11 findPeaks (http://homer.ucsd.edu/); the edges of these enriched regions were intersected with the circle-supporting reads. The threshold for circle detection was then determined empirically based on a positive control set of circular DNAs from bulk sequencing data. Only enriched regions intersected by at least two circle-supporting reads were classified as circular regions.

## Quality-controlled filtering of scCircle-seq data

To evaluate adequate enrichment of circular DNA, we used coverage over mtDNA as the internal control. Cells with fewer than ten reads per base pair sequence-read depth over mtDNA or fewer than 85% genomic bases captured in mtDNA were omitted from further analyses. Cutoff values were chosen based on maximal read depth values detected in endonuclease controls (with PmeI; Supplementary Fig. 1c). For all downstream analyses, we only considered sequencing data from cells digested with exonuclease for 5 days. Because mtDNA is not present in nuclei, we filtered single-nucleus Circle-seq data only based on RNA quality control.

## Recurrence analysis from scCircle-seq data

Read counts from putative circles were quantified using bedtools multicov (https://bedtools.readthedocs.io) from single-cell BAM files in 100-kb bins across all canonical chromosomes from genome assembly GRCh37/hg19. Counts were normalized to sequencing depth in each cell and each bin was marked positive if it contained circle read enrichment with P < 0.05 compared with the background read distribution. Bins were then classified into three groups based on genomic coordinates: (1) ecDNA if the region overlapped the amplicon assembled from the bulk sequencing data; (2) chrM; and (3) all other sites. Recurrence was then analyzed by plotting the fraction of cells containing a detected circle in each of the three categories.

## Phasing of SNPs in scCircle-seq data

Reference phasing was used to assign each SNP to one of the two alleles based on bulk WGS data. Then, single cells were genotyped to compare if the same allele was gained in all of them. For this analysis, we used the known SNPs identified by the 1000 Genomes Project[58] and extracted coverage and nucleotide counts for each annotated position. In regions with allelic imbalance, like the high copy number gains at ecDNA loci, the B-allele frequency of a heterozygous SNP is significantly different from 0.5. Hence, we could assign each SNP in these regions to either the gained or non-gained allele. We then also genotyped all single cells at each known SNP location and visualized the resulting B-allele frequency values while keeping the allele assignment from the bulk WGS data.

## Relative copy number estimation (log₂ coverage)

The average coverage over all annotated genes was calculated and genes were split into amplicon and non-amplicon genes based on whether their genomic location overlapped with the identified ecDNA regions per cell. The coverage of all amplicon genes was normalized by the background coverage, that is, the winsorized mean coverage of all non-amplicon genes. A winsorized mean was chosen to account for the fact that the identification of ecDNA regions might have been incomplete; thus, the top and bottom 5% of values were removed from the background coverage. The resulting values were $\log_2$-transformed and used as a proxy for ecDNA copy number.

## Identification of SVs in scCircle-seq data

The SV calling for scCircle-seq was done using lumpy-sv[55] (v.0.2.14) and SvABA(v.1.1.0). To our knowledge, no dedicated SV caller for single-cell DNA data is available. However, because of high copy numbers of ecDNA, bulk methods work.

## Identification of SVs in WGS bulk data and merged scCircle-seq data

SAMtools[59] (v.1.11) was used to merge all alignment files of the same cell line into one pseudobulk alignment. To achieve a coverage closer to standard bulk sequencing, the resulting BAM file was subsequently downsampled to 10% of its original size using SAMtools. The identification of SVs in WGS and merged scCircle-seq data for the TR14 and CHP-212 cell lines was accomplished using lumpy-sv[60] (v.0.3.1) and SvABA[61] (v.1.1.0), both with standard parameters. The preprocessing of the BAM files, which included lower size (<20 bp) and lower quality reads (MAPQ < 5) filtering, as well as supporting read counts and VAF calculations, was performed using SAMtools[59] (v.1.10). All the analysis steps were completed using the GRCh37/hg19 reference genome. The identification and counts of reads supporting the SV breakpoints were performed considering split and abnormally mapped reads and filtering out duplicated reads and secondary alignments.

## Identification of SNVs in bulk WGS data and merged scCircle-seq data

To ensure compatibility with standard mitochondrial variation reporting[62], each single-cell sequencing sample was realigned to GRCh37/

hg19 with a substituted revised Cambridge Reference Sequence mitochondrial reference (GenBank no. NC_012920) using BWA-MEM[63] (v.0.7.17). Duplicate reads were removed using Picard (v.2.23.8). GATK4/Mutect2[64] (v.4.1.9.0) with default parameters was used to call variants in whole-genome bulk and merged scCircle-seq sequencing data (pseudobulk). Only variants on canonical chromosomes (including chrM) and passing GATK4/FilterMutectCalls were retained and subsequently filtered for the regions previously reconstructed for the respective cell lines (Fig. 3a) using bcftools filter with flag -r.

## Identification of SNVs in mtDNA

For mitochondrial SNV identification in single cells, we applied a custom pipeline consisting of GATK4/Mutect2 (ref. 64) (v.4.1.9.0) in mitochondria mode and Mutserve[65] (v.2.0.0-rc12), a variant caller optimized to detect heteroplasmic sites in mitochondrial sequencing data, with default parameters. First, variants were called by both callers for each single cell separately. Variants were then filtered in a two-step process: (1) variants were only retained if they have been called in at least two samples by the same caller; and (2) remaining variants were only kept if they were called by both callers. Variants labeled 'blacklist' by Mutserve were removed. To infer the allele frequency for each variant in the final set, each single cell was then subjected to genotyping using alleleCount (v.4.0.2) (https://github.com/cancerit/alleleCount). Only reads uniquely mapping to the mitochondrial reference and with a mapping quality ≥ 30 were kept. For each called alternate allele $b$ at position $x$, the allele frequency (AF) was calculated as:

$$AF_{x,b} = \frac{(\text{read count})_{x,b}}{\text{read depth}_x}$$

The resulting single-cell $x$ variant AF matrix was further filtered manually and separately for each cell line. Single cells with fewer than three variants and variants with a maximum column allele frequency < 5%, mean AF (MAF) > 30% and MAF < 0.1% for CHP-212 as well as MAF > 30% and MAF < 0.1% for TR14 were considered uninformative for clustering and removed based on spot checking.

Heatmap visualization of the filtered single-cell $x$ variant AF matrix was generated using the R package ComplexHeatmap[66] (v.2.6.2). Hierarchical clustering was then applied to the single cells using the R package hclust with the agglomeration method parameter 'complete'. Phylogenetic trees were rendered using the R package dendextend (v.1.15.2).

## Microhomology detection

Microhomology analysis was performed using NCBI BLAST (https://blast.ncbi.nlm.nih.gov/Blast.cgi) with the following parameters: blastn -task megablast -word_size = 4 -evalue = 1 -outfmt '6 qseqid length evalue' -subject_besthit -reward = 1 -penalty = -2. These parameters look for a minimum microhomology length of 4 bp, and the standard reward and penalty values for nucleotide match and mismatch. In addition, we only considered significant results with an Expect value < 1. To evaluate the presence of microhomology around the circular DNA junctions, we generated files that include 100 bp around the start and end of the circle (50 bp inside the circular DNA and 50 bp of linear DNA). To be able to perform this analysis, we filtered out all the circles with a length <100 bp. Then, we compared the sequences for each start and end pair (one circle junction), evaluating and retrieving microhomologous sequences around the circular junction. This analysis was repeated for each individual circle in the CHP-212 and TR14 cell lines.

## Quality control filtering and clustering of scRNA-seq data

Cells and nuclei were loaded into Seurat[67] (v.4.10); features that were detected in at least three cells were included. Subsequently, cells with 5,000 or more features in cell lines and 2,000 features in T cells and nuclei were selected for further analysis. Cells or nuclei with high expression of mitochondrial genes (>15% in single cells and >2.5% in

nuclei) were also excluded. Data were normalized with a scale factor of 10.000 and scaled using default ScaleData settings. To account for gene length and total read count in each cell, the Smart-seq2 data were normalized using transcripts per million; then, a pseudocount of one was added and natural-log transformation was applied. The first four principal components were significant; therefore, the first five principal components were used for FindNeighbors and RunUMAP to capture as much variation as possible as recommended by the Seurat authors. The resolution for FindClusters was set to 0.5.

## Cell cycle analyses in scRNA-seq data

Cell cycle phase was assigned to single cells based on the expression of G2/M and S phase markers using the Seurat CellCycleScoring function.

## Single-cell differential expression analysis

Very small circular DNAs were defined as circles shorter than 3 kb. To calculate the relative number of this subtype of small circular DNAs per cell, the number of <3 kb circular DNAs was divided by the total number of circles in a cell. The cells were ranked by their relative number and grouped by taking the top and bottom 40% of the ranked list, defined as 'high' and 'low', respectively. Logarithmic fold change of gene expression between the two groups was calculated using the FindMarkers function in the Seurat R package[67] (v.4.10) without logarithmic fold change threshold and a minimum detection rate per gene of 0.05. The R package clusterProfiler[68] (v.4.0.5) was used to perform unsupervised GSEA of gene ontology terms using gseGO and including gene sets with at least three genes and a maximum of 800 genes.

## Correlation of scCircle-seq and scRNA-seq coverage

Coverage of ecDNA amplicon regions in the scCircle-seq and scRNA-seq BAM files was calculated with bamCoverage[55] using CPM normalization. Correlation between Circle-seq and RNA-seq coverage was analyzed by fitting a linear model.

## Identification of fusion genes

The single-cell, paired-end, RNA-seq FASTQ files were merged (96 cells for TR14 and 192 cells for CHP-212). The obtained merged data were aligned with STAR[69] (v.2.7.9a) to the reference decoy GRCh37/hs37d5, using the GENCODE 19 gene annotation, allowing for chimeric alignment (--chimOutType WithinBAM SoftClip). To call and visualize fusion genes, Arriba[70] (v.2.1.0) was applied, with the custom parameters -F 150 -U 700. The final confident call set included only fusions with (1) total coverage across the breakpoint ≥ 50× and (2) ≥30% of the mapped reads being split or discordant reads. Only fusion genes in the proximity (±10 Mb) of the amplicon boundaries were considered for the downstream analysis.

## ecDNA amplicon reconstruction

We used the amplicon reconstructions provided by Helmsauer et al.[28] for CHP-212 and Hung et al.[23] for TR14. Briefly, these reconstructions were obtained by organizing a filtered set of Illumina WGS (CHP-212) and Nanopore WGS (TR14) SV calls as genome graphs using gGnome[71] (v.0.1) (genomic intervals as nodes and reference or SVs as edges). Then, circular paths through these graphs were identified that included the amplified oncogenes and could account for the major copy number steps observed in the respective cell line. For the two patients added to the study, patient no. 1 and patient no. 2, shallow whole-genome Nanopore data were generated as described by Helmsauer et al.[28]. Basecalling, read filtering (NanoFilt −l300), mapping and SV calling were performed as described previously in the Methods ('Nanopore scCircle-seq data processing'). For ecDNA reconstruction, a set of confident SV calls was compiled (variant AF > 0.2 and supporting reads ≥ 50×). As for CHP-212 and TR14, a genome graph was built using gGnome[61] (v.0.1) and manually curated. To check amplicon structure correctness for the patient samples, in silico-simulated Nanopore reads were sampled from the

reconstructed amplicon using an adapted version of PBSIM2 (ref. 72) (https://github.com/madagiurgiu25/pbsim2) and preprocessed as the original patient samples. Lastly, the SV profiles between original samples and in silico simulation were compared. All reconstructed amplicons were visualized using gTrack (v.0.1.0; https://github.com/mskilab/gTrack), including the GRCh37/hg19 reference genome and GENCODE 19 track.

### ecDNAs co-occurrence analysis in TR14 single cells

We used the circle classification algorithm described previously to define circular DNA-enriched regions in single cells. For each single cell, we defined whether the circular DNA-enriched regions overlapped the ecDNA amplicon (*MYNC*, *CDK4*, *MDM2*) assembled from TR14 bulk sequencing data using the function findOverlaps from the R package GenomicRanges[73] (v.1.44.0). Presence or absence of overlap was defined for each of the three *MYNC*, *CDK4*, *MDM2* ecDNAs independently, excluding the amplicon regions shared by *MYCN* and *CDK4* ecDNAs.

### Statistics and reproducibility

No statistical method was used to predetermine sample size. No data were excluded from the analyses. Experiments were not randomized and the investigators were not blinded to allocation during the experiments and outcome assessment. The FISH experiments were performed once per cell line and primary tumor.

### Reporting summary

Further information on research design is available in the Nature Portfolio Reporting Summary linked to this article.

## Data availability

The sequencing data generated in this study are available at the European Genome-phenome Archive under accession no. EGAS00001007026. The ChIP–seq narrowPeak and bigwig files were downloaded from https://data.cyverse.org/dav-anon/iplant/home/konstantin/helmsaueretal/. All other data are available from the corresponding author upon reasonable request. Source data are provided with this paper.

## Code availability

The data analysis code associated with this publication can be found at https://github.com/henssen-lab/scEC-T-seq.

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

## Acknowledgements

A.G.H. is supported by the Deutsche Forschungsgemeinschaft (DFG) (grant no. 398299703). This project has received funding from the European Research Council under the European Union's Horizon 2020 Research and Innovation Programme (grant no. 949172). A.G.H. is supported by the Deutsche Krebshilfe Mildred Scheel Professorship program no. 70114107. This project was supported by the Berlin Institute of Health (BIH). Computation was performed on the HPC for Research cluster of the BIH. The project that gave rise to these results

received the support of a fellowship from 'la Caixa' Foundation (no. 100010434). The fellowship code is LCF/BQ/EU20/11810051. E.R.-F. is supported by the Alexander von Humboldt Foundation. R.X. is supported by Deutsche Krebshilfe. R.P.K. is a BIH Visiting Professor funded by the Stiftung Charité. M.C.S. is funded by the DFG-funded Research Training Group 2424/CompCancer. R.F.S. is a Professor at the Cancer Research Center Cologne Essen funded by the Ministry of Culture and Science of the State of North Rhine-Westphalia. This work was partially funded by the German Ministry for Education and Research as BIFOLD (Berlin Institute for the Foundations of Learning and Data) (refs. 01IS18025A and 01IS18037A). This work was delivered as part of the eDyNAmiC team supported by the Cancer Grand Challenges partnership funded by Cancer Research UK (H.Y.C. no. CGCATF-2021/100012, A.G.H. no. CGCATF-2021/100017) and the National Cancer Institute (H.Y.C. no. OT2CA278688, A.G.H. no. OT2CA278644).

## Author contributions

R.C.G., T.C., R.P.K. and A.G.H. contributed to study design, and data collection and interpretation. R.C.G., T.C. and K.K. performed the single-cell experiments. R.P.K., R.C.G. and K.Haase performed the analysis of the scCircle-seq data and WGS. E.R.-F. and M.G. performed the SV analysis of the single-cell and WGS data. E.V. and M.C.S. performed the scRNA-seq data analysis. M.G. performed the fusion gene detection analyses. R.X. performed the SNV analyses in the single-cell and WGS data. L.B. performed and analyzed the FISH. K.Helmsauer and M.G. performed the amplicon reconstruction analyses. M.E.S. performed the ChIP-seq. K.Helmsauer performed the ChIP–seq analyses. H.D.G., K.S., Y.B., M.L. and K.L.H. performed the experiments and contributed to the data analysis. S.M., H.Y.C., H.E.D., S.S., A.E., J.H.S. and R.F.S. contributed to study design. R.C.G., R.P.K. and A.G.H. led study design, performed the data analysis and wrote the manuscript, to which all authors contributed.

## Competing interests

R.P.K. and A.G.H. are founders of Econic Biosciences Ltd.

## Additional information

**Correspondence and requests for materials** should be addressed to Anton G. Henssen.

| | |
|---|---|

# Reporting Summary

## Statistics

For all statistical analyses, confirm that the following items are present in the figure legend, table legend, main text, or Methods section.

| n/a | Confirmed | |
|---|---|---|
| ☐ | ☒ | The exact sample size (*n*) for each experimental group/condition, given as a discrete number and unit of measurement |
| ☒ | ☐ | A statement on whether measurements were taken from distinct samples or whether the same sample was measured repeatedly |
| ☐ | ☒ | The statistical test(s) used AND whether they are one- or two-sided<br>*Only common tests should be described solely by name; describe more complex techniques in the Methods section.* |
| ☒ | ☐ | A description of all covariates tested |
| ☒ | ☐ | A description of any assumptions or corrections, such as tests of normality and adjustment for multiple comparisons |
| ☐ | ☒ | A full description of the statistical parameters including central tendency (e.g. means) or other basic estimates (e.g. regression coefficient) AND variation (e.g. standard deviation) or associated estimates of uncertainty (e.g. confidence intervals) |
| ☐ | ☒ | For null hypothesis testing, the test statistic (e.g. *F*, *t*, *r*) with confidence intervals, effect sizes, degrees of freedom and *P* value noted<br>*Give P values as exact values whenever suitable.* |
| ☒ | ☐ | For Bayesian analysis, information on the choice of priors and Markov chain Monte Carlo settings |
| ☒ | ☐ | For hierarchical and complex designs, identification of the appropriate level for tests and full reporting of outcomes |
| ☐ | ☒ | Estimates of effect sizes (e.g. Cohen's *d*, Pearson's *r*), indicating how they were calculated |

*Our web collection on statistics for biologists contains articles on many of the points above.*

## Software and code

Policy information about availability of computer code

| Data collection | FACSAria Fusion flow cytometer (Biosciences) |
|---|---|
| Data analysis | FIJI (version 2.1.0), Trim Galore (version 0.6.4), Bwa mem (version 0.7.17 and version 0.7.15), Hisat2 (version 2.2.1), rsem (version 1.3.1), Guppy (version 5.0.14), NanoFilt (version 2.8.0), ngmlr (version 0.2.7), Sniffles(version 1.0.12) , deepTools (version 3.5.1) , FASTQC (version 0.11.8), Picard (version 2.20.4 and 2.16.0 and 2.23.8), BBMap (version 38.58), regioneR(version 1.24.0), Homer (version 4.11), Lumpy-sv(version 0.2.14 and 0.3.1), SvABA (version 1.1.0), Samtools (version 1.11 and 1.10), GATK4/Mutect2(version 4.1.9.0), Mutserve(version 2.0.0-rc12), AlleleCounter (version 4.0.2), ComplexHeatmap(version 2.6.2), dendextend (version 1.15.2), Seurat (4.10), clusterProfiler (4.0.5), STAR (version 2.7.9a), Arriba (version2.1.0), gGnome (version 0.1), gTrack(0.1.0), Genomic Ranges (1.44.0), ggplot (version 3.4.1) . Data analysis code associated with this publication can be found here:  https://github.com/henssen-lab/scEC-T-seq |

For manuscripts utilizing custom algorithms or software that are central to the research but not yet described in published literature, software must be made available to editors and reviewers. We strongly encourage code deposition in a community repository (e.g. GitHub). See the Nature Portfolio guidelines for submitting code & software for further information.

## Data

Policy information about availability of data

All manuscripts must include a data availability statement. This statement should provide the following information, where applicable:
- Accession codes, unique identifiers, or web links for publicly available datasets
- A description of any restrictions on data availability
- For clinical datasets or third party data, please ensure that the statement adheres to our policy

Sequencing data generated in this study are available at the European Genome-phenome Archive (EGA) under the accesion number: EGAS00001007026. ChIP-seq

# Field-specific reporting

Please select the one below that is the best fit for your research. If you are not sure, read the appropriate sections before making your selection.

☒ Life sciences  ☐ Behavioural & social sciences  ☐ Ecological, evolutionary & environmental sciences

For a reference copy of the document with all sections, see nature.com/documents/nr-reporting-summary-flat.pdf

# Life sciences study design

All studies must disclose on these points even when the disclosure is negative.

| | |
|---|---|
| Sample size | No statistical methods were used to predetermine sample size. Sample size was decided based on multiple factors including: <br> (1) The cell numbers used in previous publications from similar low-throughput single cell genomic methods, which allowed them to detect statistically significant differences between conditions. <br> (2) Cost <br> (3) Tissue samples availability |
| Data exclusions | No data was excluded for analyses |
| Replication | Biological replicates across two different cell lines and tissue samples showed consistent and reproducible experimental results. To exclude bacht effects and ensure reproducibility, we included and processed in parallel multiple experimental conditions and, when possible multiple cell lines, in different 96-well plates. FISH experiments were done once per cell line and primary tumor. |
| Randomization | There was no subject randomization in this study. |
| Blinding | Patients were deidentified and assigned a study-specific identification number. Investigators were not blinded to allocation during experiments and outcome assessment. |

# Reporting for specific materials, systems and methods

We require information from authors about some types of materials, experimental systems and methods used in many studies. Here, indicate whether each material, system or method listed is relevant to your study. If you are not sure if a list item applies to your research, read the appropriate section before selecting a response.

## Materials & experimental systems

| n/a | Involved in the study |
|---|---|
| ☐ | ☒ Antibodies |
| ☐ | ☒ Eukaryotic cell lines |
| ☒ | ☐ Palaeontology and archaeology |
| ☒ | ☐ Animals and other organisms |
| ☐ | ☒ Human research participants |
| ☒ | ☐ Clinical data |
| ☒ | ☐ Dual use research of concern |

## Methods

| n/a | Involved in the study |
|---|---|
| ☐ | ☒ ChIP-seq |
| ☐ | ☒ Flow cytometry |
| ☒ | ☐ MRI-based neuroimaging |

## Antibodies

| | |
|---|---|
| Antibodies used | Anti-H3K27me3 polyclonal antibody (Sigma-Aldrich, cat. # 07-449, lot #2382150 , dilution 1:480 ) |
| Validation | Anti-H3K27me3 (Sigma-Aldrich, cat. # 07-449, lot #2382150) has been validated for ChIP-seq by the manufacturer (https://www.abcam.com/histone-h3-tri-methyl-k27-antibody-epr18607-chip-grade-ab192985.html) |

## Eukaryotic cell lines

Policy information about cell lines

| | |
|---|---|
| Cell line source(s) | Human tumor cell lines were obtained from the American Type Culture Collection (CHP-212; ATCC; Manassas, VA, USA) or were kindly provided by J. J. Molenaar (TR14; Princess Máxima Center for Pediatric Oncology, Utrecht, Netherlands). |
| Authentication | The identity of all cell lines was verified by STR genotyping (Genetica DNA Laboratories and IDEXX BioResearch) |

| Mycoplasma contamination | Absence of Mycoplasma sp. contamination was determined with a Lonza MycoAlert system (Lonza) |
|---|---|
| Commonly misidentified lines<br>(See ICLAC register) | The cell lines included in this study (CHP-212 and TR14) are confirmed neuroblastoma cell lines. No commonly misidentified cell lines were used in this study. |

# Human research participants

Policy information about studies involving human research participants

| Population characteristics | This study comprised the analyses of tumor and blood samples of patients diagnosed with neuroblastoma between 1991 and 2022. Age or gender is irrelevant to this study. |
|---|---|
| Recruitment | Patients were registered and treated according to the trial protocols of the German Society of Pediatric Oncology and Hematology (GPOH). This study was conducted in accordance with the World Medical Association Declaration of Helsinki (2013) and good clinical practice; informed consent was obtained from all patients or their guardians. No bias in recrutment is relevant to this study. |
| Ethics oversight | The collection and use of patient specimens was approved by the institutional review boards of Charité-Universitätsmedizin Berlin and the Medical Faculty, University of Cologne. |

Note that full information on the approval of the study protocol must also be provided in the manuscript.

# ChIP-seq

## Data deposition

☒ Confirm that both raw and final processed data have been deposited in a public database such as GEO.

☒ Confirm that you have deposited or provided access to graph files (e.g. BED files) for the called peaks.

| Data access links<br>*May remain private before publication.* | ATAC-seq and H3K4me1, H3K27ac ChIP-seq narrowpeak and bigwig files from Helmsauer et al. were downloaded at https://data.cyverse.org/dav-anon/iplant/home/konstantin/helmsaueretal/. Sequencing raw data from Helmsauer et al are available at the Sequence Read Archive under accession PRJNA622577.<br><br>Generated H3K27me3 ChIP-seq raw data, as well as big wig and narrow peak bed files, are available at the European Genome-phenome Archive (EGA) under the accesion number: EGAS00001007026. |
|---|---|
| Files in database submission | The following data files are available for ATAC-seq, H3K4me1, H3K27ac and H3K27me3 under the data access links provided above:<br>- fastaq files<br>- bw files (hg19)<br>- narrow peaks bed files |
| Genome browser session<br>(e.g. UCSC) | No longer applicable |

## Methodology

| Replicates | This study is mainly based on published ChIP-seq data which had not been acquired in replicates. |
|---|---|
| Sequencing depth | ChIP-seq: 25M 75bp single-end reads |
| Antibodies | Anti-H3K27me3 polyclonal antibody |
| Peak calling parameters | MACS2 (2.1.2) with default parameters |
| Data quality | Data was quality controlled using RPC, NPC and composite plots over housekeeping genes. |
| Software | FASTQC 0.11.8, BBMap 38.58, BWA-MEM 0.7.15, Picard 2.20.4, Deeptools 3.3.0, MACS2 2.1.2 |

# Flow Cytometry

## Plots

Confirm that:

☒ The axis labels state the marker and fluorochrome used (e.g. CD4-FITC).

☒ The axis scales are clearly visible. Include numbers along axes only for bottom left plot of group (a 'group' is an analysis of identical markers).

☒ All plots are contour plots with outliers or pseudocolor plots.

☒ A numerical value for number of cells or percentage (with statistics) is provided.

## Methodology

| | |
|---|---|
| Sample preparation | CHP-212 or TR14 cells in culture were trypsinized, washed once with 1xPBS and resuspended in 1xPBS and Propidium Iodide (PI) for sorting. Peripherial blood mononuclear cells (PBMCs) were isolated using density gradient centrifugation with Ficoll resuspended and Propidium Iodide PI in 1xPBS for sorting. . PBMCs suspensions were additionally stained with a 1:400 dilution of anti-human CD3. Nuclei suspensions were stained with DAPI |
| Instrument | FACSAria Fusion flow cytometer (Biosciences) |
| Software | FlowJo 10.7.1 |
| Cell population abundance | Cell percentages are shown in Supplementary figures 1a,b and 14a,b. CD3+ DAP- live, T-cell population abundance was 6% in patient #3 derived PBMCs and 14.5% in patient #4 derived PBMCs. |
| Gating strategy | In all cases, events were separated from debris using Forward scatter (FSC) vs. side scatter (SSC). To separate live from dead cells Forward scatter (FSC) vs. propidium iodide (PI) was used in cell lines (TR14 and CHP-212). CD3+ and DAPI- PBMCs were sorted. In the case of nuclei, DAPI+ nuclei were sorted. |

☒ Tick this box to confirm that a figure exemplifying the gating strategy is provided in the Supplementary Information.

