## [Peer Review File · Nature Genetics]

Peer Review Information

Manuscript Title: Parallel sequencing of extrachromosomal circular DNAs and transcriptomes in single cancer cells

Corresponding author name(s): Anton Henssen

Reviewer Comments & Decisions:

Decision Letter, initial version:

8th Mar 2022

Dear Dr Henssen,

First of all, I am so sorry that it has taken so long to return this decision to you. Thank you so much for your patience.

Your Technical Report, "Parallel sequencing of extrachromosomal circular DNAs and transcriptomes in single cancer cells" has now been seen by 4 referees. Please note that Reviewers #3 and #4 reviewed the paper together and uploaded the same report. You will see from their comments copied below that while they find your work of considerable potential interest, they have raised quite substantial concerns that must be addressed. In light of these comments, we cannot accept the manuscript for publication, but would be very interested in considering a revised version that addresses these serious concerns.

We hope you will find the referees' comments useful as you decide how to proceed. If you wish to submit a substantially revised manuscript, please bear in mind that we will be reluctant to approach the referees again in the absence of major revisions.

To guide the scope of the revisions, the editors discuss the referee reports in detail within the team, including with the chief editor, with a view to identifying key priorities that should be addressed in revision and sometimes overruling referee requests that are deemed beyond the scope of the current study. In this case, we think that the lack of validation on primary tumour material (which has been raised by all reviewers) will need to be addressed as a priority. However, we think that all the reviewer comments will need to be addressed experimentally (ideally, where appropriate) or textually. We appreciate that there might be some aspects of the method (such as throughput, raised by Reviewers #3 and #4) that you are unlikely to be able to change at this late stage. However, the onus is on you to convince these reviewers that your method still adds significant value despite its potential limitations. In this respect, we'd suggest that you showcase its potential by addressing reviewer queries pertaining to ecDNA biology that have been raised throughout. We have a high bar in this

space and we would expect to see wholesale support from the reviewers in the next round before agreeing to proceed further.

Please do not hesitate to get in touch if you would like to discuss these issues further.

If you choose to revise your manuscript taking into account all reviewer and editor comments, please highlight all changes in the manuscript text file. At this stage we will need you to upload a copy of the manuscript in MS Word .docx or similar editable format.

*2) If you have not done so already please begin to revise your manuscript so that it conforms to our Technical Report format instructions, available [here](http://www.nature.com/ng/authors/article_types/index.html). Refer also to any guidelines provided in this letter.

[redacted]

If you wish to submit a suitably revised manuscript we would hope to receive it within 6 months. If you cannot send it within this time, please let us know. We will be happy to consider your revision so long as nothing similar has been accepted for publication at Nature Genetics or published elsewhere. Should your manuscript be substantially delayed without notifying us in advance and your article is eventually published, the received date would be that of the revised, not the original, version.

Please do not hesitate to contact me if you have any questions or would like to discuss the required

revisions further.

Thank you for the opportunity to review your work.

Sincerely,

Safia Danovi
Editor
Nature Genetics

Referee expertise:

Referee #1: ecDNAs

Referee #2: single cell methods, cancer genomics

Referee #3 and #4: single cell methods, cancer genomics (these reviewers reviewed the paper together and have uploaded the same report)

Reviewers' Comments:

Reviewer #1:

Remarks to the Author:

In the manuscript "Parallel sequencing of extrachromosomal circular DNAs and transcripts in single cancer cells", Gonzalez and colleagues describe a novel approach to simultaneously measure gene expression and characterize ecDNAs at the single cell level.

The manuscript is well written and the topic is of great interest to the scientific community and to the broad readership of Nature Genetics. The relevant scientific literature is cited, and the statistical analyses used are, as far as I can judge, appropriate.

The experiments performed are clearly described and their results are generally consistent with the conclusions reached by the authors.

I have two major issues that I would like to see addressed before publication, as I am convinced they would significantly strengthen the manuscript.

1) All the analyses described are performed on two cancer cell lines known to have abundant ecDNAs. It would be very useful to include the analysis of at least one non-transformed primary cell type and (less critical) of cells isolated from a primary tumor. Including normal cells would be important to exclude that the wide range of cell-private circular DNAs observed by authors are not an artifact of the methodology used to enrich for and sequence ecDNAs.

2) The second aspect that I would like to see clarified concerns the larger, oncogene-containing, shared ecDNAs. From figure 1 it is not clear how much this class of circ DNA is enriched by the extensive exonuclease treatment used in the method. Judging from extended figure 3, it seems that the vast majority of ecDNAs sequenced are between 10 and 100 Kbp in size, with only a very small fraction larger than this. This is not surprising, as one would imagine that the larger the ecDNA is, the more likely it is to be accidentally fragmented during isolation, and consequently less likely to survive the exonuclease treatment. As these larger ecDNAs are the ones of most interest to the cancer research community, some clarification on this aspect would be beneficial. For example, it would be useful to split fig 1e by ecDNA class/size.

3) Since the authors can map SNV in individual cells, it would be useful to determine whether the non-private ecDNAs (those with MYCN, MDM2, CDK4) come from only one of the two parental chromosomes (as one may expect)

Minor points:

4) Line 177-178: "Considering that scEC&T-seq was able to recurrently detect megabase-sized ecDNAs 177 containing MYCN, CDK4, MDM2, in both CHP-212 and TR14 (Fig. 2b)" needs to be clarified to avoid confusion since there are no Mdm2-containing ecDNAs in CHP-212 cells (see also figure 4d).

5) line 250: there is no Fig. 3e-h

Reviewer #2:

Remarks to the Author:

This study describes scEC&T-seq, a method for the parallel isolation and sequencing of extrachromosomal circular DNAs (ecDNA) and full-length mRNA from the same single cells. ecDNA is widespread in cancer, is an important marker for advanced tumours, considered a driving force in tumour evolution and an important source of tumour heterogeneity. A method to study this important form of structural variation driver in single cancer cells has previously been missing, and this single-cell multi-omic protocol is hence an exciting and important contribution to the field. The authors apply this method to investigate the structure and transcriptional effects of ecDNAs using cancer cell lines. As expected, oncogene-containing ecDNA elements are clonally present in cancer cells, whereas such that do not contain oncogenes appear only subclonally. This suggests that the primary mechanism by which ecDNA contributes to positive selection of tumours is via gene amplification rather than other mechanisms (e.g. repurposing of enhancers) - which is perhaps not unexpected, but goes somewhat against recent reports that primarily emphasised the gene regulatory functions of cellular ecDNA. Additionally, the authors can demonstrate further utility of their method by using ecDNA akin to mitochondrial DNA for lineage tracing. This is a succinct paper, which I read with a lot of interest. But

the paper does not appear to be quite as strong in terms of biological application and “biological take home messages”. It does demonstrate an important (and seemingly functioning) single cell technology. The points below should allow the authors to further expand on key biological questions that may be addressable using scEC&T-seq, and also to clarify the limitations of the technology.

- The key question raised in recent literature is whether ecDNA not containing amplified oncogenes could act as a driver in cancer. The authors would have the ideal method to test this experimentally, but have not touched on this important outstanding question really. They are currently somewhat narrow in their application to two cancer cell lines, and should consider expanding this further, ideally to a couple of primary samples, to allow them to address this key question.

- Can the authors more formally test whether there is evidence for enhancer elements under positive selection when in ecDNAs (rather than merely oncogene copy-number being under positive selection), and/or whether amplified enhancer are associated with any globally measured cellular transcriptional effect? If oncogene copy-number are the main/only cancer driver in ecDNA, this would suggest that mainstream protocols (copy-number profiling) might largely suffice to search for ecDNA-contained driver genes in future studies.

- Since the method offers insights into the rearrangement structure of ecDNAs, I would have expected additional analyses that shed light on how ecDNAs are formed and how they might be subject to further (potentially ongoing) rearrangement evolution. The authors have access to a vast pool of ecDNAs including such not rising to high clonal frequencies. Do the authors observe microhomology at the rearrangement breakpoints indicative of the action of particular error-prone DNA repair processes? Do they observe evidence of circular recombination as a mechanism to increase ecDNA heterogeneity during cancer evolution as proposed by Rosswog et al. (PMID: 34782764)?

- Limitations of the approach should be comprehensively provided. Particularly, the authors should make clear that somatic rearrangements not existing in an extrachromosomal state are not captured. It focuses on an important form of somatic structural variation on the one hand, but is also narrow in not enabling to characterise other forms on the other hand (including ecDNAs that reintegrated). Could scEC&T-seq be paired with other single-cell technologies, such as WGA-based techniques (which again is narrowly focused on the subset of structural variations amplifying or leading to loss of genes) or with single-cell tri channel processing (scTRIP) to potentially increase the spectrum of somatic variation covered in future single-cell studies?

Additional comment:

- Can the authors clarify any ecDNA size constraints that their method might be having, either at the lower or higher end of the scDNA size spectrum?

Reviewer #3:

Remarks to the Author:

In this manuscript, González and colleagues developed a method to sequence extrachromosomal circular DNAs (ecDNAs) and profile transcriptomics in two neuroblastoma cancer cell lines (CHP-212 and TR14) at single-cell resolution. After FACS sorting to separate the single live cancer cell into 96-well plates, the DNA and polyadenylated RNA from each single cell were separated by using magnetic

beads coupled with Oligo dT primers. The DNA part was mixed with exonuclease for 1 or 5 days to remove linear DNA. The rest circular DNA was amplified by rolling circle amplification with phi29 DNA polymerase for sequencing and profiling the ecDNAs. The RNA was processed with Smart-seq2 approach to profile the single cell transcriptomics, which represents a first-generation (older) approach to profile the transcriptome, whereas most new methods (eg. 10X) use poly-A priming and high-throughput systems (eg. nanowells or microdroplets). This research suggests scEC&T can simultaneously sequence ecDNAs and mRNAs in the same single cancer cell. The results suggest this method can quantify the extrachromosomal circular DNAs as well as detect the complex structure of multi-fragmented ecDNAs and single nucleotide variants (SNVs) on ecDNAs and mitochondrial DNAs. In addition, the transcriptome analysis can be used to measure the expression and fusion structural variants of the oncogene on the ecDNAs. While the authors showed that their method can achieve DNA&RNA profiling of ecDNAs, there are significant limitations in the procedure (low-throughput, extensive laboratory protocols) which are very similar to previously published methods such as G&T-seq (excluding the digestion of linear DNA) and are unlikely to be widely adopted by the research community. Furthermore, this method is only tested with cancer neuroblastoma cell lines while its performance on detecting ecDNAs from human or mouse tumor tissue specimens is unclear. The multiple displacement amplification (MDA) can amplify the DNA effectively but highly unevenly, which may not allow the number of ecDNAs to be quantified accurately. The authors will require additional evidence to indicate scEC&T can accurately quantify the number of ecDNAs with oncogenes.

Major comments:

1. The authors developed scEC&T method to simultaneously sequence ecDNAs and mRNAs in the two neuroblastoma cancer cells lines. However, many techniques work well in cell lines but fail in tumor samples (eg. human tissue or mouse). The authors need to show that scEC&T can also effectively detect and profile ecDNAs in tumor specimens from cancer patients. I suggest the authors also provide the single cell ecDNAs sequencing data from 2-3 patients, which may significantly strengthen this manuscript.
2. The scEC&T method is very low-throughput and requires extensive laboratory work to separate the RNA from the DNA using beads, and perform digestion of the linear DNA prior to running the Phi29 based WGA procedures. The DNA-RNA sequencing approach is very similar to G&T-seq and other methods that have already been published in the field. Therefore, it is unlikely that this method can be used on a high-throughout system (eg. nanowells, microdroplets) and will be widely used by the research community.
3. Multiple displacement amplification (MDA) can effectively amplify DNA but in a highly uneven manner. The authors use the data generated by MDA to infer the number of circular DNAs (Figure 2a), is this number accurately indicates the ecDNAs number in the cells? In Figure 2b, what is the copy number of MYCN in CHP-212 cells and what is the copy number of MYCN, CDK4, and MDM2 in TR14 cells? Can scEC&T method quantify the number of the ecDNAs with these oncogenes? The authors need to show the copy number of these genes from scEC&T data and compare to the copy number of these genes profiled from either bulk sequencing or single cell whole genome sequencing (eg. using a tagmentation approach which is more accurate to profile CNVs).
4. In Fig 3a and 3b, what is the unit of the read coverage over the ecDNA region in single cells? Can the authors provide the copy number or the copy ratio instead? If so, can the authors do clustering analysis to subgroup these single cells with the ecDNAs? In addition, the authors made pseudo-bulk data from scEC&T data to detect the ecDNA-associated structural variants. I was also wondering at

least how many single cells need to be used to detect the structural variant? Can the authors detect the structural variant by just using data from one single-cell, or is the technique limited to consensus level measurements? Did the authors need ecDNA reconstruction from bulk Illumina sequencing data or Nanopore sequencing data as reference to detect the structural variants in the scEC&T dataset? It will be more powerful if scEC&T method can directly profile the complex structure of the ecDNAs. Can the authors perform long-read sequencing (exp. Nanopore sequencing) to sequence the ecDNAs at single-cell level after circle rolling amplification?

5. The authors mentioned that recurrent oncogene-containing ecDNAs harbor MYCN, CDK4, and MDM2 may confer fitness advantage. Authors found that CHP-212 cancer cells harbor 1 ecDNA harbor MYCN and TR14 cancer cells carry 3 independent ecDNAs harbor MYCN, CDK4, and MDM2 separately. It could be very interesting to the readers to learn the copy number differences of these ecDNAs in the cancer cell population. For example, will the cells carry more copies of these ecDNAs have a higher proportion in the cell population? In addition, in Fig 4a-c, only 25 TR14 cells (digested by exonuclease for 5 days) were used to show the intercellular difference. Can the authors also utilize the other 41 TR14 cells data (digested by exonuclease for 1 day) to do the same analysis to show intercellular differences of ecDNAs?

Minor comments:

1. Please provide how many million reads were used to sequence ecDNAs in each cancer cell and how many million reads were used to profile transcriptome in each cancer cell.
2. In Extended Data Figure 7b, it is difficult to read this heatmap for lineage tracing. Can the author improve the clustering of the single-cell mtDNA sequencing data? Can the author use mtDNA SNVs to build a phylogenetic tree?
3. Line251: Figure legend (Fig. 3e-h) seems wrong.
4. Line428: what is volume used in washing step?
5. Line 456: is this reaction performed in 96 well plate?
6. Line 484: please remove "!" after Galore.

Reviewer #4:

Remarks to the Author:

In this manuscript, González and colleagues developed a method to sequence extrachromosomal circular DNAs (ecDNAs) and profile transcriptomics in two neuroblastoma cancer cell lines (CHP-212 and TR14) at single-cell resolution. After FACS sorting to separate the single live cancer cell into 96-well plates, the DNA and polyadenylated RNA from each single cell were separated by using magnetic beads coupled with Oligo dT primers. The DNA part was mixed with exonuclease for 1 or 5 days to remove linear DNA. The rest circular DNA was amplified by rolling circle amplification with phi29 DNA polymerase for sequencing and profiling the ecDNAs. The RNA was processed with Smart-seq2 approach to profile the single cell transcriptomics, which represents a first-generation (older) approach to profile the transcriptome, whereas most new methods (eg. 10X) use poly-A priming and high-throughput systems (eg. nanowells or microdroplets). This research suggests scEC&T can simultaneously sequence ecDNAs and mRNAs in the same single cancer cell. The results suggest this method can quantify the extrachromosomal circular DNAs as well as detect the complex structure of multi-fragmented ecDNAs and single nucleotide variants (SNVs) on ecDNAs and mitochondrial DNAs. In addition, the transcriptome analysis can be used to measure the expression and fusion structural variants of the oncogene on the ecDNAs. While the authors showed that their method can achieve

DNA&RNA profiling of ecDNAs, there are significant limitations in the procedure (low-throughput, extensive laboratory protocols) which are very similar to previously published methods such as G&T-seq (excluding the digestion of linear DNA) and are unlikely to be widely adopted by the research community. Furthermore, this method is only tested with cancer neuroblastoma cell lines while its performance on detecting ecDNAs from human or mouse tumor tissue specimens is unclear. The multiple displacement amplification (MDA) can amplify the DNA effectively but highly unevenly, which may not allow the number of ecDNAs to be quantified accurately. The authors will require additional evidence to indicate scEC&T can accurately quantify the number of ecDNAs with oncogenes.

Major comments:

1. The authors developed scEC&T method to simultaneously sequence ecDNAs and mRNAs in the two neuroblastoma cancer cells lines. However, many techniques work well in cell lines but fail in tumor samples (eg. human tissue or mouse). The authors need to show that scEC&T can also effectively detect and profile ecDNAs in tumor specimens from cancer patients. I suggest the authors also provide the single cell ecDNAs sequencing data from 2-3 patients, which may significantly strengthen this manuscript.

2. The scEC&T method is very low-throughput and requires extensive laboratory work to separate the RNA from the DNA using beads, and perform digestion of the linear DNA prior to running the Phi29 based WGA procedures. The DNA-RNA sequencing approach is very similar to G&T-seq and other methods that have already been published in the field. Therefore, it is unlikely that this method can be used on a high-throughput system (eg. nanowell, microdroplets) and will be widely used by the research community.

3. Multiple displacement amplification (MDA) can effectively amplify DNA but in a highly uneven manner. The authors use the data generated by MDA to infer the number of circular DNAs (Figure 2a), is this number accurately indicates the ecDNAs number in the cells? In Figure 2b, what is the copy number of MYCN in CHP-212 cells and what is the copy number of MYCN, CDK4, and MDM2 in TR14 cells? Can scEC&T method quantify the number of the ecDNAs with these oncogenes? The authors need to show the copy number of these genes from scEC&T data and compare to the copy number of these genes profiled from either bulk sequencing or single cell whole genome sequencing (eg. using a tagmentation approach which is more accurate to profile CNVs).

4. In Fig 3a and 3b, what is the unit of the read coverage over the ecDNA region in single cells? Can the authors provide the copy number or the copy ratio instead? If so, can the authors do clustering analysis to subgroup these single cells with the ecDNAs? In addition, the authors made pseudo-bulk data from scEC&T data to detect the ecDNA-associated structural variants. I was also wondering at least how many single cells need to be used to detect the structural variant? Can the authors detect the structural variant by just using data from one single-cell, or is the technique limited to consensus level measurements? Did the authors need ecDNA reconstruction from bulk Illumina sequencing data or Nanopore sequencing data as reference to detect the structural variants in the scEC&T dataset? It will be more powerful if scEC&T method can directly profile the complex structure of the ecDNAs. Can the authors perform long-read sequencing (exp. Nanopore sequencing) to sequence the ecDNAs at single-cell level after circle rolling amplification?

5. The authors mentioned that recurrent oncogene-containing ecDNAs harbor MYCN, CDK4, and MDM2 may confer fitness advantage. Authors found that CHP-212 cancer cells harbor 1 ecDNA harbor MYCN and TR14 cancer cells carry 3 independent ecDNAs harbor MYCN, CDK4, and MDM2 separately. It

could be very interesting to the readers to learn the copy number differences of these ecDNAs in the cancer cell population. For example, will the cells carry more copies of these ecDNAs have a higher proportion in the cell population? In addition, in Fig 4a-c, only 25 TR14 cells (digested by exonuclease for 5 days) were used to show the intercellular difference. Can the authors also utilize the other 41 TR14 cells data (digested by exonuclease for 1 day) to do the same analysis to show intercellular differences of ecDNAs?

Minor comments:

1. Please provide how many million reads were used to sequence ecDNAs in each cancer cell and how many million reads were used to profile transcriptome in each cancer cell.
2. In Extended Data Figure 7b, it is difficult to read this heatmap for lineage tracing. Can the author improve the clustering of the single-cell mtDNA sequencing data? Can the author use mtDNA SNVs to build a phylogenetic tree?
3. Line251: Figure legend (Fig. 3e-h) seems wrong.
4. Line428: what is volume used in washing step?
5. Line 456: is this reaction performed in 96 well plate?
6. Line 484: please remove "!" after Galore.

Author Rebuttal to Initial comments

Point-by-point response to reviewers' comment to manuscript NG-TR59148 Editor's

remarks:

To guide the scope of the revisions, the editors discuss the referee reports in detail within the team, including with the chief editor, with a view to identifying key priorities that should be addressed in revision and sometimes overruling referee requests that are deemed beyond the scope of the current study. In this case, we think that the lack of validation on primary tumor material (which has been raised by all reviewers) will need to be addressed as a priority. However, we think that all the reviewer comments will need to be addressed experimentally (ideally, where appropriate) or textually. We appreciate that there might be some aspects of the method (such as throughput, raised by Reviewers #3 and #4) that you are unlikely to be able to change at this late stage. However, the onus is on you to convince these reviewers that your method still adds significant value despite its potential limitations. In this respect, we'd suggest that you showcase its potential by addressing reviewer queries pertaining to ecDNA biology that have been raised throughout. We have a high bar in this space, and we would expect to see wholesale support from the reviewers in the next round before agreeing to proceed further.

Response: We thank the editors for giving us the opportunity to revise this manuscript. Following the editor's suggestions, we have addressed all reviewers' comments experimentally and/or textually (see point by point response below). Notably, we have now validated our method in primary tumors and normal tissue. Using this new data we now showcase the potential of our new method by addressing important open questions, eg. about ecDNA structural dynamics.

Reviewer #1 (Remarks to the Author):

In the manuscript "Parallel sequencing of extrachromosomal circular DNAs and transcriptomes in single cancer cells", Gonzalez and colleagues describe a novel approach to simultaneously measure gene expression and characterize ecDNAs at the single cell level. The manuscript is well written and the topic is of great interest to the scientific community and to the broad readership of Nature Genetics. The relevant scientific literature is cited, and the statistical analyses used are, as far as I can judge, appropriate. The experiments performed are clearly described and their results are generally consistent with the conclusions reached by the authors.

Response: We thank the reviewer for the supportive comments.

I have two major issues that I would like to see addressed before publication, as I am convinced, they would significantly strengthen the manuscript.

1) All the analyses described are performed on two cancer cell lines known to have abundant ecDNAs. It would be very useful to include the analysis of at least one non-transformed primary cell type and (less critical) of cells isolated from a primary tumor. Including normal cells would be important to exclude that the wide range of cell-private circular DNAs observed by authors are not an artifact of the methodology used to enrich for and sequence ecDNAs.

Response: We thank the reviewer for their valuable suggestions. We now applied our method to single nuclei isolated from two independent primary neuroblastomas and single non-transformed, non-ecDNA containing T-cells isolated from blood samples of two different patients (see new Figs. 6 + 7, also copied below and p. 11-15, l. 334-485). No ecDNA, i.e. large oncogene-containing extrachromosomal circular DNAs, were detected in the lymphocytes, whereas tumor cells harbored such ecDNA (see. Fig. 6c), as confirmed by FISH (Extended Data Fig. 15e). Small circular DNAs, on the other hand, were also found in lymphocytes, but at significantly lower abundance (see Fig. 6b).

New Figure 6: scEC&T-seq detects extrachromosomal circular DNAs in primary neuroblastomas. **a**, Schematic diagram describing tumor and blood sample processing. **b**, Number of individual circular DNA regions normalized by library size detected in primary tumor nuclei ($n = 93$ nuclei patient #1, $n = 86$ nuclei patient #2), neuroblastoma cell line single cells ($n = 25$ TR14 cells, $n = 150$ CHP-212 cells) and non-malignant single T-cells ($n = 38$ patient #3, $n = 41$ patient #4; P -values calculated using Welch's t -test). **c**, Heat map of the genome-wide circular DNA density in neuroblastoma primary tumors and normal T-cells ($n = 93$ patient #1, green; $n = 86$ patient #2, purple; $n = 38$ patient #3, yellow; $n = 41$ patient #4, orange; bin sizes = 3 Mb).

New Figure 7: Intercellular heterogeneity in ecDNA structure reveals ecDNA structural dynamics in primary neuroblastomas. **a**, Long-read based amplicon reconstructions derived from WGS sequencing data in bulk populations and read coverage over the ecDNA fragments across single nuclei in patient #2 ($n = 86$ nuclei) as detected by long-read or short-read scEC&T-seq. Top to bottom: ecDNA amplicon reconstruction, gene annotation, read density over the ecDNA region in bulk long-read nanopore WGS data, read density over the ecDNA region in merged single cells, coverage over the ecDNA region in single cells (rows) as detected by long-read or short-read scEC&T-seq. **b**, Heatmap of total number of reads (log-scaled) in 500 bp windows over the identified deletion in patient 2's ecDNA structure across single nuclei. **c**, Exemplary genome tracks of the three identified subclone variants of patient #2. Log-scaled total read density is shown in blue and split read density is shown in grey. **d**, Detection of SVs supporting the consensus bulk ecDNA structure in 8 single cells representing the 3 identified subclones (≥ 1 read supporting the SV is shown in grey, 0 reads supporting the SV is shown in white). **e**, ecDNA reconstructions of the identified ecDNA variants in patient #2 single nuclei. **f**, Schematic interpretation of the ecDNA structural dynamics predicted in patient #2 based on the identified ecDNA structural subclonal variants in scEC&T-seq data. The position of the *MYCN* oncogene and its local enhancers (e1-e5) in each ecDNA variant is indicated.

2) The second aspect that I would like to see clarified concerns the larger, oncogene-containing, shared ecDNAs. From Figure 1 it is not clear how much this class of circular DNA is enriched by the extensive exonuclease treatment used in the method. Judging from Extended Data Figure 3 (now Extended Data Fig. 4a), it seems that the vast majority of ecDNAs sequenced are between 10 and 100 Kbp in size, with only a very small fraction larger than this. This is not surprising, as one would imagine that the larger the ecDNA is, the more likely it is to be accidentally fragmented during isolation, and consequently less likely to survive the exonuclease treatment. As these larger ecDNAs are the ones of most interest to the cancer research community, some clarification on this aspect would be beneficial. For example, it would be useful to split Fig. 1e by ecDNA class/size.

Response: We thank the reviewer for this very valuable suggestion. Our method is able to capture megabase size ecDNA up to 1.2 Mb (see Extended Data Fig. 4a and l. 173 p. 6). To address the reviewer's comment, we added two new supplementary figures, one for each circular DNA class (large ecDNAs vs other smaller circular DNAs), see Extended Data Fig. 1e, f, also copied below. Indeed, enrichment is most significant for smaller circular DNA regions, especially after 5-day exonuclease digestion (Extended Data Fig. 1f). Significant enrichment of reads mapping to large ecDNAs (i.e. larger than 100kb) is observed after 1-day exonuclease digestion. The enrichment is not as pronounced after prolonged 5-day exonuclease digestion, in line with the reviewer's idea that large ecDNA may be fragmented and less stable in the presence of exonuclease compared to smaller circular DNAs (Extended Data Fig. 1e). We added a sentence to the results section (p. 5, l. 121-127) describing this difference in enrichment and the potential limitations.

Revised Extended Data Fig. 1 (partial): Quality control of single cell Circle-seq data. **e**, Fraction of sequencing reads mapping to ecDNA regions in each experimental condition in CHP-212 and TR14 cells. Welch's *t*-test was used among indicated conditions. **f**, Fraction of sequencing reads mapping to other circular DNA regions (excluding ecDNA regions and mtDNA) in each experimental condition in CHP-212 and TR14 cells. Welch's *t*-test was used among indicated conditions.

3) Since the authors can map SNV in individual cells, it would be useful to determine whether the non-private ecDNAs (those with *MYCN*, *MDM2*, *CDK4*) come from only one of the two parental chromosomes (as one may expect).

Response: This is an interesting question, which we also addressed in our previous work (Koche et al, Nature Genetics 2020). Analogous to that approach, we used reference phasing to assign each SNP to one of the two alleles based on bulk sequencing data and then genotyped every single cell to compare if the same allele is gained in all of them. For this analysis we used the known SNPs identified by the 1000 genomes project and extracted coverage and nucleotide counts for each annotated position. In regions with allelic imbalance, like the high copy number gains at ecDNA loci, the B-allele frequency (BAF) of a heterozygous SNP is significantly different from 0.5. Hence, we could assign each SNP in these regions to either the gained or non-gained allele (see Extended Data Fig. 9a, also copied below). We then also genotyped all single cells at each known SNP location and visualized the resulting BAF values while keeping the allele assignment from

the bulk data. Thus, a difference in gained vs non-gained allele would be immediately visible in the figures by an inversion of the two colors. All single cells show the same allele as gained (see Extended Data Fig. 9b, also copied below). The results were consistent across single cell data. So, phasing of SNPs suggests that ecDNA are of mono- allelic origin.

New Extended Data Fig. 9: Phasing of SNPs in ecDNA loci in scEC&T-seq data indicates ecDNAs are of mono-allelic origin. a, Reference phasing of *MYCN*, *CDK4* and *MDM2* ecDNA loci in bulk WGS data. Shown is the raw sequencing coverage (top) and the B-allele frequency (BAF; bottom) of known SNPs based on the 1000 genomes annotation. SNPs in regions of high-level amplifications can be very clearly assigned to the gained or non-gained allele based on BAF. **b,** Genotyped *MYCN* ecDNA locus in scCircle-seq CHP-212 and TR14 sequencing data (6 exemplary cells in each case are shown). Shown is the B-allele frequency (BAF) of known SNPs based on the 1000 genomes annotation. SNPs that have been reference phased based on bulk sequencing data are colored the same as in (a). All cells show the same allele as gained in concordance with the bulk data analysis.

Minor points:

4) Line 177-178: "Considering that scEC&T-seq was able to recurrently detect megabase-sized ecDNAs containing MYCN, CDK4, MDM2, in both CHP-212 and TR14 (Fig. 2b)" needs to be clarified to avoid confusion since there are no Mdm2-containing ecDNAs in CHP-212 cells (see also Fig. 4d).

Response: We thank the reviewer for pointing this out. We have modified the text accordingly. "*Considering that scEC&T-seq was able to recurrently detect megabase- sized ecDNAs in both CHP-212 and TR14 (Fig. 2b)...*". (p. 7, l. 195-196)

5) Line 250: there is no Fig. 3e-h

Response: We thank the reviewer for pointing this out and have corrected "*Fig. 3 e-h*" to "*Fig. 4 e-h*", as we were referring to this figure.

Reviewer #2 (Remarks to the Author):

This study describes scEC&T-seq, a method for the parallel isolation and sequencing of extrachromosomal circular DNAs (ecDNA) and full-length mRNA from the same single cells. ecDNA is widespread in cancer, is an important marker for advanced tumors, considered a driving force in tumor evolution and an important source of tumor heterogeneity. A method to study this important form of structural variation driver in single cancer cells has previously been missing, and this single-cell multi-omic protocol is hence an exciting and important contribution to the field. The authors apply this method to investigate the structure and transcriptional effects of ecDNAs using cancer cell lines. As expected, oncogene-containing ecDNA elements are clonally present in cancer cells, whereas such that do not contain oncogenes appear only subclonally. This suggests that the primary mechanism by which ecDNA contributes to positive selection of tumors is via gene amplification rather than other mechanisms (e.g. repurposing of enhancers) - which is perhaps not unexpected, but goes somewhat against recent reports that primarily emphasized the gene regulatory functions of cellular ecDNA. Additionally, the authors can demonstrate further utility of their method by using ecDNA akin to mitochondrial DNA for lineage tracing. This is a succinct paper, which I read with a lot of interest. But the paper does not appear to be quite as strong in terms of biological application and "biological take home messages". It does demonstrate an important (and seemingly functioning) single cell technology. The points below should allow the authors to further expand on key biological questions that may be addressable using scEC&T-seq, and also to clarify the limitations of the technology.

1) The key question raised in recent literature is whether ecDNA not containing amplified oncogenes could act as a driver in cancer. The authors would have the ideal method to test this experimentally but

have not touched on this important outstanding question really. They are currently somewhat narrow in their application to two cancer cell lines, and should consider expanding this further, ideally to a couple of primary samples, to allow them to address this key question.

Response: We thank the reviewer for their appreciation of our method and the important comments, which have helped us improve our manuscript. To address the reviewer's suggestions, we have now applied our method to two primary neuroblastomas (see new Figs. 6 + 7, also copied above and p. 11-15, l. 334-485) and have added an entire section about the inclusion of enhancers on ecDNA (see l. 448-485, 14-15). This not only confirms that our method is applicable to primary tumors, but also enabled us to investigate new features of structural ecDNA heterogeneity, which may help us better understand ecDNA structural dynamics/evolution (see. new Fig. 7, also copied above). Even though some of the smaller extrachromosomal circular DNAs contained regulatory elements (see new Extended Data Fig. 17c, and below in response to comment #2), we did not identify any recurrent elements in cell lines or primary neuroblastomas that did not contain oncogenes, which is in contrast to previous reports suggesting that such ecDNAs may exist^{1,2} (see l. 448-485, p. 14-15). This does not exclude that such ecDNA exist in other tumor entities or samples, but due to the lack of recurrent ecDNAs containing only regulatory elements, we are unable to address this outstanding and important question. We have added a section to the discussion about this open question, which in principle could be addressed using our method when applied to tumors that contain such ecDNAs (see p.17, l. 545-549).

2) Can the authors more formally test whether there is evidence for enhancer elements under positive selection when in ecDNAs (rather than merely oncogene copy-number being under positive selection), and/or whether amplified enhancers are associated with any globally measured cellular transcriptional effect? If oncogene copy-number are the main/only cancer driver in ecDNA, this would suggest that mainstream protocols (copy-number profiling) might largely suffice to search for ecDNA-contained driver genes in future studies.

Response: As mentioned in our response to reviewer's comment #1, we did not identify any recurrent ecDNAs lacking oncogenes and only containing enhancer elements in our cell lines and tumors. We now, however, describe the structural heterogeneity of a *MYCN*-containing ecDNA in a primary neuroblastoma in detail, which confirms that enhancers are recurrently included in ecDNAs (see p. 14-15, l. 448-485 and Fig. 7f, also copied above, and Extended Data Fig. 17a, also copied below). In order to identify circular DNAs only harboring regulatory elements and address the reviewer's question, we analyzed H3K4me1, H3K27Ac, H3K27me3 ChIP-seq and ATAC-seq data of CHP-212 to identify regulatory elements (eg. enhancers). We identified a large number of cell private smaller circular DNAs, some of which contained regions of regulatory elements (see new Extended Data Fig. 17b and below). None of these circular DNAs, however, were recurrently detected in more than one single cells, suggesting that these circular DNAs likely did not confer any selective advantage or that these circular DNAs were not maintained/propagated, due to yet unknown structural limitations. Additionally, we tested whether certain regulatory elements were enriched on circular DNAs by comparing the expected ChIP-seq signal within such circular DNAs vs. the

observed ChIP-seq signal (Extended Data Fig. 13 e,f,g,h, copied below). Interestingly, the signal for all measured histone marks were significantly lower than expected in regions from which circular DNAs originated. Thus, no recurrent ecDNAs only harboring regulatory elements were detected in the neuroblastoma cancer samples. Our results, however, suggest that our method can be useful to perform the suggested analyses in cancer cells harboring recurrent non-oncogene containing ecDNAs.

New Extended Data Fig. 17: MYCN local enhancers are recurrently included in ecDNA in neuroblastoma. **a**, Genome tracks of long-read nanopore WGS data showing co-amplification in ecDNA of MYCN local enhancers (e1- e5) with the MYCN proto-oncogene in neuroblastoma cell lines (CHP-212 and TR14) and primary tumors (patient #1 and #2). **b**, Overlap of non-recurrent extrachromosomal circular DNAs with chromatin marks. Top to bottom: 5 exemplary genome tracks of 5 different CHP-212 single cells (Log-scaled total read density in blue and circle edge read density in grey); H3K27ac ChIP-seq (dark green); H3K4me1 ChIP-seq (light green); H3K27me3 ChIP-seq (red); ATAC-seq (black); CTCF ChIP-seq (purple).

New Extended Data Fig. 13 (partial): Chromatin marks and chromatin accessibility in extrachromosomal circular DNAs

in single cells. e-h, Mean ChIP-seq or ATAC-seq signal across all detected circular DNA regions in all CHP-212 single cells (red) and randomized signal (black): ATAC-seq (e, $p = 0.001$); H3K4me1 ChIP-seq (f, $p = 0.001$); H3K27ac ChIP-seq (g, $p = 0.001$); H3K27me3 ChIP-seq (h, $p = 0.001$).

3) Since the method offers insights into the rearrangement structure of ecDNAs, I would have expected additional analyses that shed light on how ecDNAs are formed and how they might be subject to further (potentially ongoing) rearrangement evolution. The authors have access to a vast pool of ecDNAs including such not rising to high clonal frequencies. Do the authors observe microhomology at the rearrangement breakpoints indicative of the action of particular error-prone DNA repair processes? Do they observe evidence of circular recombination as a mechanism to increase ecDNA heterogeneity during cancer evolution as proposed by Rosswog et al. (PMID: 34782764)?

Response: We appreciate the reviewer's suggestion and agree that our method is suitable to address these important questions. To address the question of structural ecDNA heterogeneity and evolution, we applied our method to primary neuroblastomas (see Fig. 6 and 7, also copied and described above). Indeed, we were able to characterize the structure and intercellular structural heterogeneity of an ecDNA in a neuroblastoma (Fig. 7). This revealed that ecDNA can structurally differ between single cancers cells. Moreover, such structural heterogeneity helped infer ecDNA structural evolution and identify footprints of circular recombination (Fig. 7e,f and p. 13-14, l. 399-446), in line with the studies referenced by the reviewer. Thus, our method is able to detect ecDNA structural heterogeneity and is useful to investigate ecDNA evolution.

As suggested by the reviewer, we also measured the presence of microhomology at circular junctions by comparing 100 bp sequences around each of the breakpoints using BLAST. We observed microhomology in 76.6% of all extrachromosomal circular DNAs, with a mean length of 10 bp, suggesting microhomology-mediated repair mechanisms may contribute to circularization (see new Extended Data Figure 12, also copied below and p. 10, l. 306-309). Serendipitously, while addressing reviewer's comment #2, we observed an unexpected enrichment of CTCF binding sites at circular DNA breakpoints, as measured using CTCF ChIP-seq, both in single and bulk CHP-212 neuroblastoma cells (see new Extended Data Fig. 13a-d and below). Even though this will need further testing, this suggests that sites of CTCF-mediated loop extrusion may be prone to break and form extrachromosomal circular DNAs in cancer cells. Thus, our method may help foster the discovery of new mechanisms of ecDNA generation. We thank the reviewer for their suggestion, as without it we would not have made this potentially exciting discovery.

New Extended Data Fig. 12: Microhomology detection at circular DNA breakpoints. a, Length distribution of microhomologies in CHP-212 and TR14 single cells.

New Extended Data Fig. 13 (partial): Chromatin marks and chromatin accessibility in extrachromosomal circular DNAs in single cells. a, b, Fraction of circular DNA edge regions overlapping with CTCF ChIP-seq peaks in CHP-212 single cells (a) and in bulk CHP-212 Circle-seq (b). Overlap shown in red and randomized overlap in dark.

c,d, Mean CTCF signal around the edges (dashed line) of circular DNA regions detected in all CHP-212 single cells (c) and in bulk CHP-212 Circle-seq (d).

4) Limitations of the approach should be comprehensively provided. Particularly, the authors should make clear that somatic rearrangements not existing in an extrachromosomal state are not captured. It focuses on an important form of somatic structural variation on the one hand but is also narrow in not enabling to characterize other forms on the other hand (including ecDNAs that reintegrated). Could scEC&T-seq be paired with other single-cell technologies, such as WGA- based techniques (which again is narrowly focused on the subset of structural variations amplifying or leading to loss of genes) or with single-cell tri channel processing (scTRIP) to potentially increase the spectrum of somatic variation covered in future single-cell studies?

Response: We agree with the reviewer that our method is inherently unable to detect variants on linear DNA. We now discuss this limitation in our manuscript and agree that combining our method with other approaches may allow a more comprehensive analysis of genomic variants in single cancer cells, eg. ecDNA re-integration (see p. 16 , l. 510- 518).

Additional comment:

5) Can the authors clarify any ecDNA size constraints that their method might be having, either at the lower or higher end of the scDNA size spectrum?

Response: We identify ecDNA elements from 30 bp to 1.2 Mb in length, which we now specifically mention in the text (p. 6, l. 173, Extended Data Fig. 4a). As discussed above (see comment #2 to reviewer #1), larger ecDNAs may be more fragile and more prone to fragmentation and therefore be digested by exonucleases during our protocol and small circular DNAs may be more efficiently amplified using rolling circle amplification. However, the fact that we do detect large ecDNA suggests that the presence of ecDNA in high copy numbers makes it more likely that some ecDNAs remain intact and survive exonuclease digestion. We now mention this and discuss it in the revised manuscript (p. 5, l. 121-127).

Reviewer #3 and #4 (Remarks to the Author):

In this manuscript, González and colleagues developed a method to sequence extrachromosomal circular DNAs (ecDNAs) and profile transcriptomics in two neuroblastoma cancer cell lines (CHP-212 and TR14) at single cell resolution. After FACS sorting to separate the single live cancer cell into 96-well plates, the DNA and polyadenylated RNA from each single cell were separated by using magnetic beads coupled with Oligo dT primers. The DNA part was mixed with exonuclease for 1 or 5 days to remove linear DNA. The rest circular DNA was amplified by rolling circle amplification with phi29 DNA polymerase for sequencing and profiling the ecDNAs. The RNA was processed with Smart-seq2 approach to profile the single cell transcriptomics, which represents a first-generation (older) approach to profile the transcriptome, whereas most new methods (eg. 10X) use poly-A priming and high-throughput systems (eg. nanowells or microdroplets). This research suggests scEC&T can simultaneously sequence ecDNAs and mRNAs in the same single cancer cell. The results suggest this method can quantify the extrachromosomal circular DNAs as well as detect the complex structure of multi-fragmented ecDNAs and single nucleotide variants (SNVs) on ecDNAs and mitochondrial DNAs. In addition, the transcriptome analysis can be used to measure the expression and fusion structural variants of the oncogene on the ecDNAs. While the authors showed that their method can achieve DNA&RNA profiling of ecDNAs, there are significant limitations in the procedure (low- throughput, extensive laboratory protocols) which are very similar to previously published methods such as G&T-seq (excluding the digestion of linear DNA) and are unlikely to be widely adopted by the research community. Furthermore, this method is only tested with cancer neuroblastoma cell lines while its performance on detecting ecDNAs from human or mouse tumor tissue specimens is unclear. The multiple displacement amplification (MDA) can amplify the DNA effectively but highly unevenly, which may not allow the number of ecDNAs to be quantified accurately. The authors will require additional evidence to indicate scEC&T can accurately quantify the number of ecDNAs with oncogenes.

Response: We thank the reviewers for their valuable comments and agree that our method has inherent limitations regarding throughput and copy number estimation. To address the reviewers' question about the applicability on tumor specimen, we have now applied our method to primary neuroblastomas and human T-cells (see response to reviewer #1, comment #1 and our responses below as well as new figures 6 and 7 and corresponding results sections on p. 11-15, l. 334-485). This has revealed some intriguing new insights on ecDNA structural heterogeneity, which may help infer how ecDNA structurally evolve in tumors (see new Figs. 6 + 7, also copied above and p. 11-15, l. 334- 485). This also highlights the biological insights that can be gained using our method.

With regards to the throughput of our method, we do not believe that throughput is a major determinant of the importance and value of a method. Many low-throughput single cell methods are of great importance to the field³ and have revealed fundamental biological insights. Higher throughput methods come with their own set of major limitations, which we believe are especially relevant in the context of ecDNA analysis. For example, the number of transcripts and/or genomic DNA fragments detected per cell is greatly reduced in most high-throughput methods. This does not permit in-depth analyses of intercellular structural ecDNA heterogeneity that can help infer ecDNA structural evolution as we now present in the revised manuscript (See new Figs. 6 + 7 and result sections p. 11-15, l. 334-485). Additionally, poly-A primer-based transcriptomes used elsewhere only retrieve very short fragments of transcripts. This does not permit the assessment of transcript variants, which do arise from rearranged ecDNAs (see Fig. 3c). To show one major advantage of our method, we have now successfully sequenced a subset of the single cells with Nanopore-based long-read sequencing (see comment #4 below). This would not have been possible for most high throughput methods, as the selection and re-analysis of a subset of individual cells is not feasible there. Being part of the newly formed eDyNAmiC Cancer Grand Challenge Team, we are currently applying our method in many different collaborative projects and different tumor entities and are convinced that we will be able to address additional major biological questions about ecDNA in these contexts. Thus, our method is already being widely used in the field of ecDNA while this manuscript is under review. We thank the reviewer for the valuable suggestions, which have helped us uncover new biological insights that serve as additional proofs-of-principles that our method can reveal very important features of ecDNA not detectable with current methods to date.

Major comments:

1) The authors developed scEC&T method to simultaneously sequence ecDNAs and mRNAs in two neuroblastoma cancer cells lines. However, many techniques work well in cell lines but fail in tumor samples (eg. human tissue or mouse). The authors need to show that scEC&T can also effectively detect and profile ecDNAs in tumor specimens from cancer patients. I suggest the authors also provide the single cell ecDNAs sequencing data from 2-3 patients, which may significantly strengthen this manuscript.

Response: We thank the reviewers for this very valuable suggestion. We have now applied our method

to two independent primary neuroblastomas and two independent T lymphocyte populations from two different patients. In all cases, scEC&T seq was successful at isolating extrachromosomal circular DNAs, detecting ecDNA variants and assessing ecDNA-driven oncogene expression (see new Fig. 6 and 7, copied above, and new Extended Data Fig. 15h,i, copied below as well as result sections p. 11-15, l. 334-485). Furthermore, we now used the information on ecDNA structural heterogeneity to infer underlying structural dynamics of ecDNA (new Fig. 7). We agree with the reviewer that the addition of this data significantly strengthens this manuscript.

Extended Data Figure 15 (partial): h, i, Pairwise comparison between ecDNA and mRNA read counts from scEC&T- seq over the reconstructed *MYCN* amplicon ecDNA region in patient #2 single nuclei (f; Pearson correlation, $P = 3.2 \times 10^{-13}$, $R = 0.69$, $n = 86$ patient #2 nuclei) and in patient #1 single nuclei (g; Pearson correlation, $P = 7.6 \times 10^{-13}$, $R = 0.66$, $n = 93$ patient #1 nuclei).

2) The scEC&T method is very low-throughput and requires extensive laboratory work to separate the RNA from the DNA using beads and perform digestion of the linear DNA prior to running the Phi29 based WGA procedures. The DNA-RNA sequencing approach is very similar to G&T-seq and other methods that have already been published in the field. Therefore, it is unlikely that this method can be used on a high-throughput system (eg. nanowells, microdroplets) and will be widely used by the research community.

Response: We agree with the reviewer that our method is of low throughput and now discuss this limitation more clearly (p. 16, l. 510-517). As mentioned in our response above, we do, however, believe that throughput is not a determinant of the importance of a method. We understand the excitement about high-throughput single cell sequencing, but would like to point out that many important biological discoveries do not depend on such throughput. We believe this to be especially true for ecDNA. This method is already being widely used in the ecDNA community, addressing the reviewer's concern about its use outside our group. In collaboration with many research groups, we have also analyzed

high-throughput droplet-based scATAC and RNA sequencing data from tumors containing ecDNA. Even though these methods can detect a subset of ecDNAs, such methods are inherently unable to distinguish ecDNA from linear amplifications. We also found that the sparsity of data from high-throughput methods prevents the in-depth analysis of ecDNA structural variants. In contrast to the identification of rare cell

types, which is one of the main advantages of high-throughput single cell sequencing, ecDNA is present in most cancer cells and can therefore be analyzed in detail using lower throughput methods. However, many currently open biological questions about ecDNA do need to be addressed in single cancer cells. For example, it is currently unclear how ecDNA structurally evolve. We now include, for the first time to our knowledge, a detailed analysis of the structural heterogeneity of ecDNA in primary neuroblastoma. This enabled us to infer ecDNA structural dynamics, which nominates circular recombination as one potential process occurring in cancer cells (see new Fig. 6 and 7, copied above as well as result sections p. 11-15, l. 334-485). Based on this proof-of-principle analysis, we are convinced that our method will provide many more important insights into such processes in future studies.

3) Multiple displacement amplification (MDA) can effectively amplify DNA but in a highly uneven manner. The authors use the data generated by MDA to infer the number of circular DNAs (Figure 2a), is this number accurately indicates the ecDNAs number in the cells? In Figure 2b, what is the copy number of *MYCN* in CHP-212 cells and what is the copy number of *MYCN*, *CDK4*, and *MDM2* in TR14 cells? Can scEC&T method quantify the number of ecDNAs with these oncogenes? The authors need to show the copy number of these genes from scEC&T data and compare to the copy number of these genes profiled from either bulk sequencing or single cell whole genome sequencing (eg. using a tagmentation approach which is more accurate to profile CNVs).

Response: We thank the reviewer for this valuable question. Firstly, we would like to clarify that the number of circular DNAs indicated in Figure 2a refers to individual circular DNA regions/elements and not to circle copy number. We have now changed the text to clarify this (p. 6, l. 169). Secondly, we not only agree with the reviewer that MDA amplifies DNA in an uneven manner, but would also like to point out that our method relies on digesting all linear DNA and on performing amplification of the remaining DNA. This lack of a linear genomic background in our method prevents absolute copy number measurement. This also prevents us from performing tagmentation, as this would linearize circular DNAs. We do, however, detect relative difference in copy number, which we believe is sufficiently informative to address open questions about ecDNA. In order to address the reviewer's comment, we have re-analyzed our data with regards to relative copy number. We calculated the average coverage over all annotated genes and split them into amplicon and non-amplicon genes based on if their genomic location overlaps with the identified ecDNA regions per cell. We then normalized the coverage of all amplicon genes by the background coverage, i.e. the windsorised mean coverage of all non-amplicon genes. We chose to use a windsorised mean to account for the fact that the identification of ecDNA regions might have been incomplete, so we remove the top and bottom 5% of values from the background coverage. The resulting values were log₂ transformed and used as a proxy for ecDNA copy number. This quantification seems to work reasonably well, as we compared the average normalized coverage for all genes located on the CHP-212 amplicon (*LPIN*, *DDX1* and *MYCN*) and compared it against

their average expression per cell and achieved a good correlation (see Unpublished Data Fig. 1a copied below). Comparing our results to absolute copy numbers from FISH analysis shows that the relative copy

number distributions are comparable (see Unpublished Data Fig. 1b,c copied below). In the absence of another background control like known diploid regions this is the best proxy for ecDNA abundance we can calculate. In general, copy number inference of focal regions remains very challenging using any of the available single cell sequencing methods. Given the sparsity of most high- throughput single cell approaches, usually binning over relatively large genomic regions is necessary to improve the signal to noise ratio. However, when trying to quantify the copy number of short focal regions like individual ecDNA segments, large bins can average out the signal and return a much lower copy number for the affected bin. Thus, our method should in principle outperform most high-throughput single cell methods regarding copy number estimation. Other methods are also unable to distinguish ecDNAs from linear amplicons, which is why they are not suitable to address many questions regarding ecDNA. Even though we agree with the reviewer that copy number estimation in single cell sequencing data is an area of ongoing research, we believe that addressing this widely accepted open question would be outside the scope of this manuscript.

Unpublished Data Fig. 1: Relative ecDNA copy number estimation using scEC&T-seq data. **a**, Pairwise correlation between the log₂ normalized average coverage over the *MYCN* ecDNA amplicon region in CHP-212 and the average expression of genes contained in the *MYCN* ecDNA amplicon in CHP-212 (Pearson correlation, $R = 0.22$, $n = 150$). **b**, Distribution of log₂ normalized *MYCN* coverage in CHP-212 single cells. **c**, Distribution of number of foci, as identified by FISH, in CHP-212 single cells.

4) In Fig 3a and 3b, what is the unit of the read coverage over the ecDNA region in single cells? Can the authors provide the copy number or the copy ratio instead? If so, can the authors do clustering analysis to subgroup these single cells with the ecDNAs?

Response: These are linear plots, therefore there is no read coverage unit in the single- cell plots as these are not quantitative but based on presence/absence of reads in a 50bp window. Again, copy ratios (eg. LogR ratios as often used in copy number callers) would only be possible if linear DNA was still detectable, which it isn't as it is digested using exonucleases.

5) In addition, the authors made pseudo-bulk data from scEC&T-seq data to detect the ecDNA-associated structural variants. I was also wondering at least how many single cells need to be used to detect the structural variant? Can the authors detect the structural variant by just using data from one single of

cell, or is the technique limited to consensus level measurements?

Response: Thank you for this valuable question. We have now performed structural variant detection directly in single cells using Illumina and Nanopore sequencing in such single cells. Indeed, we can detect a subset of the SVs in single cells without using a pseudobulk as a reference. We now include a table of SVs detected in a subset of single cells and another table listing the number of reads supporting structural variants on ecDNA in single cells (see Extended Data Table 3 and 5). We indeed are able to detect at least one SV-supporting read per ecDNA breakpoint in appr. 30% of single cells (23% of CHP-212 single cells for *MYCN* ecDNA; 44% of TR14 single cells for *MDM2* ecDNA; 28% of TR14 single cells for *MYCN* ecDNA; 24% of TR14 single cells for *CDK4* ecDNA). We now added this information to the manuscript (p. 7, l. 210-213).

6) Did the authors need ecDNA reconstruction from bulk Illumina sequencing data or Nanopore sequencing data as reference to detect the structural variants in the scEC&T dataset? It will be more powerful if scEC&T method can directly profile the complex structure of the ecDNAs. Can the authors perform long-read sequencing (exp. Nanopore sequencing) to sequence the ecDNAs at single cell level after circle rolling amplification?

Response: We thank the reviewer for this valuable suggestion, which we believe helped us improve our manuscript. Firstly, we can recover structural variants only based on scEC&T-seq without these reconstructions (see response to your comment above). We only used these reconstructions to compare the performance of scEC&T-seq to bulk sequencing (see Extended Data Table 4). According to the reviewer's suggestion, we now performed Nanopore long-read sequencing on 18 single cancer cells derived from cell lines and primary neuroblastomas. Nanopore sequencing-based detection of extrachromosomal circular DNAs nicely correlated with that of Illumina sequencing in the same single cells (Extended Data Fig. 2 a-c also below and p. 5, l. 132-135). Nanopore sequencing also allowed the detection of structural variants on ecDNA, which we believe will be useful for the characterization of ecDNA when using scEC&T seq. This approach suggested by the reviewer nicely highlights one of the major advantages of our method compared to other high-throughput droplet-based methods. In response to a new question, eg. the reviewer's question, scEC&T-seq allowed us, the investigator, to select a subset of previously sequenced cells of interests and re-sequence these using sequencing technologies of their choice (eg. Nanopore sequencing).

Extended Data Figure 2: Nanopore-based detection of extrachromosomal circular DNAs in single cells. a, Schematic of T7 endonuclease de-branching of rolling-circle amplified DNA prior to nanopore Circle-seq. **b**, Correlation of normalized read counts from Illumina and Nanopore scCircle-seq data from a subset of CHP-212 cells (Pearson correlation = 0.95, $p = 2.2 \times 10^{-16}$, each color represents a different cell and each point is a putative circle). **c**, Genome tracks comparing read coverage across the *MYCN* ecDNA amplicon regions in Illumina (blue) vs Nanopore (pink) Circle-seq data in two exemplary cells (CHP-212 and TR14).

7) The authors mentioned that recurrent oncogene-containing ecDNAs harbor *MYCN*, *CDK4*, and *MDM2* may confer fitness advantage. Authors found that CHP-212 cancer cells harbor 1 ecDNA harbor *MYCN* and TR14 cancer cells carry 3 independent ecDNAs harbor *MYCN*, *CDK4*, and *MDM2* separately. It could be very interesting to the readers to learn the copy number differences of these ecDNAs in the cancer cell population. For example, will the cells carry more copies of these ecDNAs have a higher proportion in the cell population?

Response: We thank the reviewer for this very interesting question. The reviewer is correct in their assumption. Indeed, we very recently described that cells with higher copy number are under positive selective pressure (Lange et al. Nature Genetics, 2022, PMID: 36123406). ecDNA is binomially distributed to daughter cells in each mitosis. As a result, the ecDNA copy number distributions shift to higher copy numbers over time if conferring positive selection, see manuscript cited above. Relative copy number distributions detected using our method are consistent with those observed using FISH (see Lange et al. and figure below). These distributions are consistent with positive selection as mentioned in our recent manuscript and also fits the predictions of theoretical models. We now specifically mention this publication in our manuscript and have added a section about copy number estimation (p. 8-9, l. 251-257).

New Extended Data Figure 8. Relative ecDNA abundance measured using scEC&T-seq resembles FISH-based copy number estimates. Density plots displaying the distribution of fraction of reads mapping to ecDNA regions (%) in scCircle-seq data (top) or the number of foci identified by FISH (bottom) for each ecDNA amplicon region identified in CHP-212 (*MYCN*) and TR14 (*MYCN*, *CDK4*, *MDM2*) single cells.

8) In addition, in Fig 4a-c, only 25 TR14 cells (digested by exonuclease for 5 days) were used to show the intercellular difference. Can the authors also utilize the other 41 TR14 cells data (digested by exonuclease for 1 day) to do the same analysis to show intercellular differences of ecDNAs?

Response: We thank the reviewer for the suggestion. The distribution of ecDNA combinations does not change significantly when using these additional cells (see Unpublished Data Fig. 2 below).

Unpublished Data Fig. 2. Upset plot displaying the co-occurrence of the three ecDNA amplicons identified in TR14 (*MDM2*, *CDK4*, *MYCN*) in single cells (left, only 5-day exonuclease digested cells are included, $n = 25$ TR14 cells; right, both 1-day and 5-day exonuclease digested cells are included, $n = 61$ TR14 cells)

Minor comments:

1) Please provide how many million reads were used to sequence ecDNAs in each cancer cell and how many million reads were used to profile transcriptome in each cancer cell.

Response: Approximately, 5-10 million reads (mean = 8.5 million reads per scCircle-seq library) were

used to sequence circular DNAs in each cell. We now provide the exact number of reads per library in the new Extended Table 1 (Column: total number of reads) and 2 (RNA, number of UMIs).

2) In Extended Data Figure 7b, it is difficult to read this heatmap for lineage tracing. Can the author improve the clustering of the single cell mtDNA sequencing data? Can the author use mtDNA SNVs to build a phylogenetic tree?

Response: We have now improved the analysis of lineage inference based on mtDNA sequencing data and have added an updated figure to the extended data and below (Extended Data Fig. 10a,b). We also used this to infer phylogenetic trees, see new figure Extended Data Fig. 11a,b , also shown below.

Extended Data Figure 10: scEC&T-seq enables identification of homoplasmic and heteroplasmic variants (SNVs) in

mitochondrial DNA. a, Homoplasmic mitochondrial single nucleotide variants (SNV) detected in CHP- 212 and TR14 single cells ($n = 150$ CHP-212 cells, $n = 25$ TR14 cells). Unsupervised hierarchical clustering allows for clear separation of both cell lines based on their haplogroup variants, suggesting usage for population scale phylogeny studies. Sites with read depth ≤ 10 are shown in grey. **b**, Heteroplasmic variants detected in CHP-212 single cells ($n = 150$ CHP-212 cells). Unsupervised hierarchical clustering (y-axis) suggests usage for lineage tracing exploration and applications. Sites with read depth ≤ 10 are shown in grey.

New Extended Data Figure 11: Mitochondrial heteroplasmic SNVs can be used to infer phylogeny. a,b, Phylogenetic trees inferred from heteroplasmic single-nucleotide variants identified in mitochondrial DNA in CHP- 212 (a, $n = 148$) and TR14 (b, $n = 20$) single cells.

3.Line251: Figure legend (Fig. 3e-h) seems wrong.

Response: We thank the reviewer for pointing out this error and have corrected “Fig. 3 e-h” to “Fig. 4 e-h”, as we were referring to this figure.

4.Line428: what is volume used in washing step?

Response: We thank the reviewer for pointing this out. We added this information to the methods section accordingly. “the mRNA captured beads were washed three times at room temperature in **200 μ L** of 50 mM Tris-HCl (pH 8.3), 75 mM KCl, 3 mM MgCl₂, 10 mM DTT, 0.05% Tween-20 and 0.2 \times RNase inhibitor (SUPERasin, Life Technologies)”.

5.Line 456: is this reaction performed in 96 well plate?

Response: We thank the reviewer for pointing this out and have added this information accordingly “the DNA was eluted from the beads directly into an exonuclease digestion master mix (20 units of Plasmid-Safe ATP dependent DNase (Epicentre), 1mM ATP (Epicentre), 1 \times Plasmid-Safe buffer (Epicentre)) in a 96-well plate.”

6. Line 484: please remove “!” after Galore.

Response: We have removed this in the text according to the reviewer's suggestion but would like to point out that "!" is part of the official program name.

Point-by-point response-specific reference list

1. Yi, E. *et al.* Live-cell imaging shows uneven segregation of extrachromosomal DNA elements and transcriptionally active extrachromosomal DNA hubs in cancer. *Cancer Discovery*, candisc.1376.2021 (2021).
2. Hung, K.L. *et al.* EcDNA hubs drive cooperative intermolecular oncogene expression. *bioRxiv*, 2020.11.19.390278 (2020).
3. Kashima, Y. *et al.* Single-cell sequencing techniques from individual to multiomics analyses. *Experimental & Molecular Medicine* **52**, 1419-1427 (2020).

Decision Letter, first revision:

21st Nov 2022

Dear Dr Henssen,

I hope you are well.

Your Technical Report, "Parallel sequencing of extrachromosomal circular DNAs and transcriptomes in single cancer cells" has now been seen by 4 referees. As before, Reviewers #3 and #4 reviewed the work together and have uploaded the same report.

You will see from their comments below Reviewers #1 and #2 have no further requests. However, Reviewers #3 and #4 continue to raise points about the low-throughput of the method and the determination of ecDNA copy number (CN). For the former, we require only textual changes that discuss this limitation. With respect to the latter, we agree that the measurement of ecDNA CN is an important function and we'd strongly urge you to address these comments fully. Their minor comments should also be addressed in full.

We therefore invite you to revise your manuscript taking into account all these comments. Please highlight all changes in the manuscript text file. At this stage we will need you to upload a copy of the manuscript in MS Word .docx or similar editable format.

*2) If you have not done so already please begin to revise your manuscript so that it conforms to our Technical Report format instructions, available [here](http://www.nature.com/ng/authors/article_types/index.html). Refer also to any guidelines provided in this letter.

[redacted]

We hope to receive your revised manuscript within four to eight weeks. If you cannot send it within this time, please let us know.

Sincerely,

Safia Danovi

Editor
Nature Genetics

Reviewers' Comments:

Reviewer #1:
Remarks to the Author:

The authors have satisfactorily addressed my concerns. Despite some limitations, the method seems to be robust and informative. The inclusion of data from a few primary cancers is a welcome and important addition to the manuscript.

Reviewer #2:
Remarks to the Author:

The authors have made a number of significant additions to the manuscript. These include a comprehensive description of the structural evolution of ecDNA, reporting of ecDNA structural features including CTCF site enrichment and evidence for circular recombination, addition of data from primary tumors confirming their prior data on cell lines, inclusion and analysis of long read sequence data (nanopore), and analysis of enhancer elements (pointing to a dominant role of oncogene containing ecDNAs with respect to element clonal selection). These are very relevant additions strengthening the manuscript considerably. All my prior comments have been addressed adequately. I recommend acceptance of this work.

Jan Korbel

Reviewer #3:
Remarks to the Author:

In this revised manuscript, the authors have addressed 1/3 of our major comments: 1) the application of their method to human tumor samples (addressed), 2) the lack of cell throughput in their approach (not addressed), 3) the concern about accurately estimating copy number measurements of ecDNA (not addressed). They have also provided several additional data that was related to comments from other reviewers.

They showed the inferred evolution dynamics of the ecDNAs from Patient #2 and developed and performed single-cell ecDNAs nanopore sequencing and provided concordance evidence by comparing nanopore and illumina sequencing results. They also provided evidence that the cells with higher copy number of ecDNAs may be under positive selection. They also provided results that ecDNA are of mono-allelic origin and breakpoints frequently overlap with CTCF binding sites. These data have improved the overall manuscript and addressed comments from the other reviewers, however we still have several remaining major concerns:

1) The method is very low throughput and requires an extensive laboratory protocol. The authors try to argue that high-throughput single cell sequencing methods are not needed in most cancer studies,

but this is simply not the case for most studies. This remains to be a MAJOR limitation of the method. If the publication moves forward, it must be stated explicitly in the abstract (ie, this is a 'low-throughput' single cell sequencing approach) and discussed in detail in the discussion section, and pointed out that this is a critical future direction. Additionally, the full time-scale of the protocol should be stated explicitly in the main text and pointed out as a limitation in the discussion section.

2) It remains unclear whether the current method provides any accurate copy number estimation of the ecDNAs. While the authors point out the circular DNA is amplified more accurately than linear DNA using Phi29 MDA, there is not much evidence to support this. The main issue is that by digesting their linear DNA they no longer have a genomic diploid/ploidy reference. One possible solution is to spike in reference DNA to serve as a control for estimating the ecDNA copy number (similar to ERC spike-ins for RNA). Alternatively, they could compare their data more directly to their FISH results. In their responses they show Unpublished Figure 1b and c, but did not quantify the correlation values to the ecDNA copies. This is critical data that needs to be included prominently in the main manuscript, as it indicates the accuracy of the method.

remaining minor concerns:

3) Could the authors provide the read length from the Nanopore sequencing? It will be important to show how long the reads are that can be sequenced from this method. This will be very useful to resolve the SVs especially in single-cell data with limited reads available.

4) We noticed there are a few cells have more reads detected (> 10 folds more than other cells) from supplemental table 3. Could the authors clarify or comment if this is caused by the technology amplification bias or the ecDNAs number heterogeneity?

5) In Figure 7.d, why the subclone #2 including both ecDNAs which detected or not detected SV3?

Reviewer #4:

Remarks to the Author:

In this revised manuscript, the authors have addressed 1/3 of our major comments: 1) the application of their method to human tumor samples (addressed), 2) the lack of cell throughput in their approach (not addressed), 3) the concern about accurately estimating copy number measurements of ecDNA (not addressed). They have also provided several additional data that was related to comments from other reviewers. They showed the inferred evolution dynamics of the ecDNAs from Patient #2 and developed and performed single-cell ecDNAs nanopore sequencing and provided concordance evidence by comparing nanopore and illumina sequencing results. They also provided evidence that the cells with higher copy number of ecDNAs may be under positive selection. They also provided results that ecDNA are of mono-allelic origin and breakpoints frequently overlap with CTCF binding sites. These data have improved the overall manuscript and addressed comments from the other reviewers, however we still have several remaining major concerns:

1) The method is very low throughput and requires an extensive laboratory protocol. The authors try to argue that high-throughput single cell sequencing methods are not needed in most cancer studies, but this is simply not the case for most studies. This remains to be a MAJOR limitation of the method. If the publication moves forward, it must be stated explicitly in the abstract (ie, this is a 'low-

throughput' single cell sequencing approach) and discussed in detail in the discussion section, and pointed out that this is a critical future direction. Additionally, the full time-scale of the protocol should be stated explicitly in the main text and pointed out as a limitation in the discussion section.

2) It remains unclear whether the current method provides any accurate copy number estimation of the ecDNAs. While the authors point out the circular DNA is amplified more accurately than linear DNA using Phi29 MDA, there is not much evidence to support this. The main issue is that by digesting their linear DNA they no longer have a genomic diploid/ploidy reference. One possible solution is to spike in reference DNA to serve as a control for estimating the ecDNA copy number (similar to ERC spike-ins for RNA). Alternatively, they could compare their data more directly to their FISH results. In their responses they show Unpublished Figure 1b and c, but did not quantify the correlation values to the ecDNA copies. This is critical data that needs to be included prominently in the main manuscript, as it indicates the accuracy of the method.

remaining minor concerns:

3) Could the authors provide the read length from the Nanopore sequencing? It will be important to show how long the reads can be sequenced from this method. This will be very useful to resolve the SVs especially in single-cell data with limited reads available.

4) We noticed there are a few cells have more reads detected (> 10 folds more than other cells) from supplemental table 3. Could the authors clarify or comment if this is caused by the technology amplification bias or the ecDNAs number heterogeneity?

5) In Figure7.d, why the subclone#2 including both ecDNAs which detected or not detected SV3?

Author Rebuttal, first revision:

Point-by-point response to reviewers' comment to manuscript NG-TR59148

Editor's remarks:

Your Technical Report, "Parallel sequencing of extrachromosomal circular DNAs and transcriptomes in single cancer cells" has now been seen by 4 referees. As before, Reviewers #3 and #4 reviewed the work together and have uploaded the same report. You will see from their comments below Reviewers #1 and #2 have no further requests. However, Reviewers #3 and #4 continue to raise points about the low-throughput of the method and the determination of ecDNA copy number (CN). For the former, we require only textual changes that discuss this limitation. With respect to the latter, we agree that the measurement of ecDNA CN is an important function, and we'd strongly urge you to address these comments fully. Their minor comments should also be addressed in full. We therefore invite you to revise your manuscript taking into account all these comments

Response: We thank the editors for giving us the opportunity to revise our manuscript.

Reviewer #1 (Remarks to the Author):

The authors have satisfactorily addressed my concerns. Despite some limitations, the method seems to be robust and informative. The inclusion of data from a few primary cancers is a welcome and important addition to the manuscript.

Response: We thank the reviewer for their supportive comments and for appreciating the value of our method. We also thank the reviewer for their input and constructive feedback in the previous round of revisions, which has considerably helped us improve the manuscript.

Reviewer #2 (Remarks to the Author):

The authors have made a number of significant additions to the manuscript. These include a comprehensive description of the structural evolution of ecDNA, reporting of ecDNA structural features including CTCF site enrichment and evidence for circular recombination, addition of data from primary tumors confirming their prior data on cell lines, inclusion and analysis of long read sequence data (nanopore), and analysis of enhancer elements (pointing to a dominant role of oncogene containing ecDNAs with respect to element clonal selection). These are very relevant additions strengthening the manuscript considerably. All my prior comments have been addressed adequately. I recommend acceptance of this work. (Jan Korbelt)

Response: We thank Jan Korbelt for his constructive feedback throughout the revision process, which led us to identify interesting new aspects about ecDNAs that we now include in our manuscript.

Reviewer #3 and #4 (Remarks to the Author):

In this revised manuscript, the authors have addressed 1/3 of our major comments: 1) the application of their method to human tumor samples (addressed), 2) the lack of cell throughput in their approach (not addressed), 3) the concern about accurately estimating copy number measurements of ecDNA (not addressed). They have also provided several additional data that

was related to comments from other reviewers. They showed the inferred evolution dynamics of the ecDNA from Patient #2 and developed and performed single-cell ecDNA nanopore sequencing and provided concordance evidence by comparing nanopore and illumina sequencing results. They also provided evidence that the cells with higher copy number of ecDNAs may be under positive selection. They also provided results that ecDNA are of mono-allelic origin and breakpoints frequently overlap with CTCF binding sites. These data have improved the overall manuscript and addressed comments from the other reviewers, however we still have several remaining major concerns:

1) The method is very low throughput and requires an extensive laboratory protocol. The authors try to argue that high-throughput single cell sequencing methods are not needed in most cancer studies, but this is simply not the case for most studies. This remains to be a MAJOR limitation of the method. If the publication moves forward, it must be stated explicitly in the abstract (ie, this is a ‘low-throughput’ single cell sequencing approach) and discussed in detail in the discussion section, and pointed out that this is a critical future direction. Additionally, the full time-scale of the protocol should be stated explicitly in the main text and pointed out as a limitation in the discussion section.

Response: We have now prepared a detailed protocol of our method, which we will post on *Nature* [protocol exchange](https://www.nature.com/nprot/protocolexchange) (see attachment, <https://www.nature.com/nprot/protocolexchange>) upon acceptance of this manuscript. This will allow future studies that can focus on increasing the throughput of our method. In this protocol as well as in the revised manuscript we added a sentence that describes that from start to finish our protocol takes 8 days (p. 4, l. 114-115). As suggested by the reviewer, we more directly point out this limitation in the discussion section. We specifically state that it is a “low-throughput method” which requires extensive laboratory protocols and point out scaling the throughput up as a future direction in the discussion section (p. 16, l. 512-515).

2) It remains unclear whether the current method provides any accurate copy number estimation of the ecDNAs. While the authors point out the circular DNA is amplified more accurately than linear DNA using Phi29 MDA, there is not much evidence to support this. The main issue is that by digesting their linear DNA they no longer have a genomic diploid/ploidy reference. One possible solution is to spike in reference DNA to serve as a control for estimating the ecDNA copy number (similar to ERC spike-ins for RNA). Alternatively, they could compare their data more directly to their FISH results. In their responses they show Unpublished Figure 1b and c, but did not quantify the correlation values to the ecDNA copies. This is critical data that needs to be included prominently in the main manuscript, as it indicates the accuracy of the method.

Response: We regret that this was still unclear in the revised manuscript. We would like to point out that copy number estimation of focal amplifications (eg. ecDNA) from any type of single cell data remains a considerable challenge in the field that has not yet been fully solved by any of the current computational algorithms available to date^{1,2}. We would also like to point out that even the current gold standard for single cell ecDNA copy number measurements, fluorescence in situ hybridization (FISH), does not perfectly accurately measure absolute copy number in single cells due to fact that 2D images of interphase cells only capture a fraction of ecDNA per nucleus. This is something that we and others in the field are currently heavily working on and hope to improve in the future.

We thank the reviewer for the helpful suggestions on how we could estimate our method’s performance by comparing it to FISH results. As suggested by the reviewer, we now compared our scEC&T-seq relative

copy number estimations more directly to copy numbers assessed by FISH (New Extended Data Figure 8a-d). Both the ‘log₂ ecDNA’ coverage normalized to the linear background coverage, as well as the fraction of ecDNA- specific reads distributions resembled the FISH copy number measurements (New Extended Data Figure 8e-h). Additionally, we compared our relative copy number estimates calculated from scEC&T-seq, to those obtained from other unpublished single- cell sequencing methods that were also applied to the same cell lines, e.g. single-cell genome and transcriptome sequencing (G&T, copy numbers estimated using Ginkgo (<http://qb.cshl.edu/ginkgo>)) and 10X single-cell paired ATAC and RNA-seq, a method that we and the Chang lab had used in other cell types to estimate ecDNA copy number in the past (see Hung KL. et al, Nature 2021, PMID: 34819668). Our results indicate that both scEC&T-seq and G&T-seq copy number estimates resemble each other and resemble FISH results, suggesting that our method estimates relative ecDNA copy number nearly as well as current gold standards. In contrast, single-cell paired ATAC and RNA-seq copy number distributions did not significantly resemble FISH data (Unpublished Figure 1a-d). These results highlight that, even though absolute copy numbers cannot be obtained with scEC&T-seq due to the lack of linear DNA, i.e. ploidy information, relative copy number estimations strongly resemble FISH copy number distributions and even outperform high-throughput single-cell paired ATAC and RNA- seq copy number estimates. We included a comparison of our method to the FISH copy number counts in the revised manuscript (see new Extended Data Figure 8e-h, and copied below). The presented additional datasets (G&T and scATAC&RNA) will not be included in the main manuscript as they are part of other manuscripts that are under consideration elsewhere.

Extended Data Figure 8: Relative ecDNA copy number measured using scEC&T-seq resembles FISH-based copy number estimates. a-d, Density plots displaying the scaled, mean-centered ecDNA relative copy number distributions in CHP-212 (a, *MYCN* ecDNA) and TR14 (b, *MYCN* ecDNA; c, *CDK4* ecDNA; d, *MDM2* ecDNA), as measured by *MYCN* DNA interphase FISH (red, $n = 154$ (a), $n = 232$ (b), $n = 284$ (c), $n = 65$ (d)), log₂ *MYCN* coverage in scEC&T-seq (yellow, $n = 49$ (a), $n = 15$ (b),

$n = 6$ (c), $n = 15$ (d)) and fraction of ecDNA-specific reads in scEC&T-seq (blue, $n = 150$ (a), $n = 25$ (b-d)). **e-h**, Cumulative probability of scaled, mean-centered ecDNA relative copy

number in CHP-212 (a, *MYCN* ecDNA) and TR14 (b, *MYCN* ecDNA; c, *CDK4* ecDNA; d, *MDM2* ecDNA), as measured by *MYCN* DNA interphase FISH (red, $n = 154$ (a), $n = 232$ (b), $n = 284$ (c), $n = 65$ (d)), log₂ *MYCN* coverage in scEC&T-seq (yellow, $n = 49$ (a), $n = 15$ (b), $n = 6$ (c), $n = 15$ (d)) and fraction of ecDNA-specific reads in scEC&T-seq (blue, $n = 150$ (a), $n = 25$ (b-d)). P-values were calculated by Kolmogorov-Smirnov test.

Unpublished Data Figure 1: a-b, Density plots displaying the scaled, mean-centered ecDNA relative copy number distributions in CHP-212 (a, *MYCN* ecDNA) and TR14 (b, *MYCN* ecDNA), as measured by *MYCN* DNA interphase FISH (red, $n = 154$ (a), $n = 232$ (b)), log₂ *MYCN* coverage in scEC&T-seq (yellow, $n = 49$ (a), $n = 15$ (b)), fraction of ecDNA-specific reads in scEC&T-seq (blue, $n = 150$ (a), $n = 25$ (b)), copy number estimates from G&T-seq (purple, $n = 96$ (a), $n = 96$ (b)) and copy number estimates from scATAC-RNA-seq (green, $n = 5158$ (b)). **c-d**, Cumulative probability of scaled, mean-centered ecDNA relative abundance in CHP-212 (a, *MYCN*) and TR14 (b, *MYCN*), as measured by *MYCN* DNA interphase FISH (red, $n = 154$ (a), $n = 232$ (b)), log₂ *MYCN* coverage in scEC&T-seq (yellow, $n = 49$ (a), $n = 15$ (b)), fraction of ecDNA reads in scEC&T-seq (blue, $n = 150$ (a), $n = 25$ (b)), copy number estimates from G&T-seq (purple, $n = 96$ (a), $n = 96$ (b)) and copy number estimates from scATAC-RNA-seq (green, $n = 5158$ (b)). P-values were calculated by Kolmogorov-Smirnov test.

With regards to the ability of Phi29 to more efficiently amplify circular vs. linear DNA, this has been reported not only by us, but by many others³⁻⁶, which is why usage of MDA with circular DNA is termed “rolling circle amplification”. In RCA, the polymerase Phi29 goes around a circular template molecule and reaches the primer binding site again, displacing the newly synthesized strand and continuing DNA synthesis for several rounds. However, when the DNA template is linear, the synthesis is completed at the end of the template⁶. By definition, rolling circle amplification, therefore, more efficiently amplifies

circular vs. linear DNA. This also makes the use of spike-in controls not as simple as it may seem. One would need to assess the differences in efficacy of MDA to amplify spiked in circular DNAs of different lengths and DNA content. We agree, however, that spike ins could represent an interesting strategy to overcome linear DNA digestion by serving as reference for normalization and absolute copy number estimation. However, elaborating and testing an adequate strategy to introduce DNA spike-ins in our single cell protocol to accurately estimate ecDNA copy number is technically very challenging, and we believe is out of the scope of this manuscript. ERCC spike-in controls are a well-established set of 96 polyadenylated transcripts, which span a similar sequence

length and GC content of that found in the transcriptome of eukaryotic cells⁷. Creating a similar set of stable circular DNAs of different lengths (up to megabases in length, i.e. similar size as ecDNA) would be technically very challenging. Although ERCC spike-ins can be used for quantification of RNA transcript levels, they were originally developed to assess the technical performance of different technology platforms regarding sensitivity (RNA capture) and linearity of amplification. Although some studies recommend their use for normalization, in particular those in which the global RNA levels are expected to change across experimental conditions, most studies still rely on the experimental data to estimate normalization factors. There are several criticisms to spike-in normalization, which explain why they are not being widely used for normalization in the RNA-seq field⁸⁻¹¹. Some studies argue that, in particular cases, spike-ins may behave differently to endogenous transcripts due to differences in their biophysical properties¹⁰,

e.g. unequal capture efficiencies, or library preparation effects¹¹. This may also be the case for circular DNA spike ins. Others argue that the quantity of spike-ins added to each sample is subjected to important technical variability and might be inconsistent across samples, which limits the ability to use them for normalization⁹. This would be especially challenging when scaling this down to single cell sequencing. Finally, estimating how much should be spiked-in is challenging, particularly in single cell experiments where the quantity of endogenous RNA per cell, or in our case circular DNA, is unknown and/or very low. This can result in insufficient spike-in coverage for normalization, or in excessive coverage which takes up a large proportion of reads of the library⁹. Therefore, in order to use DNA spike-in for normalization and ecDNA absolute copy number estimation, special care needs to be taken in their design, so that they resemble a cell's circular DNA content, and their performance needs to be thoroughly evaluated. Circular plasmid spike-ins have been previously used in bulk Circle-seq protocols, but only to assess the sensitivity of their method or to evaluate differences in total circular DNA content of tissue samples^{12,13}. However, the two or three plasmids chosen for these experiments only covered a size range of 7-15 kb and 3 copy number ratios (1:50:2,500). It still remains unclear whether plasmid spike-ins can be used to accurately estimate absolute copy numbers of individual extrachromosomal circular DNAs, and this is currently a work-in-progress in the field. Nevertheless, we thank the reviewer for this very elegant and thoughtful suggestion to improve ecDNA copy number estimation, which we and our collaborators will investigate in future studies.

Minor concerns:

3) Could the authors provide the read length from the Nanopore sequencing? It will be important to show how long the reads are that can be sequenced from this method. This will be very useful to resolve the SVs especially in single-cell data with limited reads available.

Response: We thank the reviewer for pointing this out. We have now included a plot showing the distribution of read length in Nanopore scCircle-seq data in Extended Data Figure 2 (d), also showed below.

Extended Data Fig. 2 (partial): Nanopore-based detection of extrachromosomal circular DNAs in single cells. d, Read length distribution of Nanopore scCircle-seq data. Individual lines represent the average across single cells grouped by sample ($n = 6$ CHP-212 cells (blue), $n = 3$ TR-14 cells (orange), $n = 4$ patient #1 nuclei (green), $n = 5$ patient #2 nuclei (red)), whereas the shade stands for 95% confidence interval.

4) We noticed there are a few cells that have more reads detected (> 10 folds more than other cells) from supplemental table 3. Could the authors clarify or comment if this is caused by the technology amplification bias or the ecDNAs number heterogeneity?

Response: Indeed, there are some cells in Supplementary Table 3, in which we identified a higher number of SV-supporting reads in some ecDNA breakpoints compared to most cells. To evaluate whether these differences were due to ecDNA number heterogeneity, we compared relative ecDNA abundance and total number of ecDNA SV-supporting reads in single cells using our scEC&T-seq data (Unpublished Figure 2, a-e). We found a positive correlation between the relative abundance of a particular ecDNA and the total number of reads supporting its SV breakpoints (Unpublished Figure 2a-e and copied below). As an example, in one TR14 cell (plate_B1, Supplementary Table 3) we identified 840 reads (>10 fold compared to the rest) supporting the SV breakpoint in the predicted *MDM2* ecDNA structure. This TR14 cell also showed the highest relative abundance of *MDM2* ecDNA in the entire TR14 population (highlighted in red in Unpublished Fig. 2c). These results indicate that the differences observed in number of SV-supporting reads in single cells is mostly due to ecDNA copy number heterogeneity and are not introduced by amplification bias.

Unpublished Figure 2: a-e, Pairwise comparison between relative ecDNA abundance as measured by fraction of ecDNA-specific reads (%), and read counts supporting SV ecDNA breakpoints in scEC&T-seq data in CHP-212 (**a**, MYCN ecDNA, Pearson correlation, $P < 2.2 \times 10^{-16}$, $R = 0.62$, $n = 150$ CHP-212 cells), patient #2 (**b**, MYCN ecDNA, Pearson correlation, $P < 2.2 \times 10^{-16}$, $R = 0.89$, $n = 86$ patient #2 nuclei) and TR14 (**c**, MDM2 ecDNA, Pearson correlation, $P = 2.2 \times 10^{-9}$, $R = 0.89$, $n = 25$ TR14 cells; **d**, MYCN ecDNA, Pearson correlation, $P = 0.11$, $R = 0.32$, $n = 25$ TR14 cells; **e**, CDK4 ecDNA, Pearson correlation, $P = 0.39$, $R = 0.18$, $n = 25$ TR14 cells).

5) In Figure 7d, why the subclone#2 including both ecDNAs which detected or not detected SV3?

Response: From the reconstruction of patient 2's ecDNA consensus amplicon from bulk WGS data, we identified 5 individual genomic fragments connected by 4 structural variants (SVs, Fig 7a). Two of those structural variants, named SV2 and SV3, depicted both breakpoints of a single 6kb deletion. From a structural point of view, SV2 cannot exist without SV3. For this reason, we considered cells exhibiting SV1+SV2 to belong to the same subclone, named subclone #2 characterized by exhibiting reads across the 6kb deletion and reads supporting SV1+SV2 and/or SV3 (Fig 7b-d). We believe that, in this particular case, the computational methods used lacked sensitivity to detect SV3. The count of supporting reads has been performed following the parameters used by SvaBa to call structural variants, implying that no secondary/supplementary alignments were considered, to minimize the detection of false positives. Although no primary alignment reads supporting SV3 were present in SV1+SV2 in cells from subclone #2, suggesting their absence, six supplementary aligned reads were detected, suggesting that SV3 might also be present in these cells but were missed by SvaBa and supporting their classification as subclone #2.

Point-by-point response-specific references

1. Mallory, X.F., Edrisi, M., Navin, N. & Nakhleh, L. Methods for copy number aberration detection from single-cell DNA-sequencing data. *Genome Biology* **21**, 208 (2020).
2. Mallory, X.F., Edrisi, M., Navin, N. & Nakhleh, L. Assessing the performance of methods for copy number aberration detection from single-cell DNA sequencing data. *PLOS Computational Biology* **16**, e1008012 (2020).
3. Liu, D., Daubendiek, S.L., Zillman, M.A., Ryan, K. & Kool, E.T. Rolling Circle DNA Synthesis: Small Circular Oligonucleotides as Efficient Templates for DNA Polymerases. *Journal of the American Chemical Society* **118**, 1587-1594 (1996).
4. Dean, F.B., Nelson, J.R., Giesler, T.L. & Lasken, R.S. Rapid amplification of plasmid and phage DNA using Phi 29 DNA polymerase and multiply-primed rolling circle amplification. *Genome Res* **11**, 1095-9 (2001).
5. Lizardi, P.M. *et al.* Mutation detection and single-molecule counting using isothermal rolling-circle amplification. *Nature Genetics* **19**, 225-232 (1998).
6. Johne, R., Müller, H., Rector, A., van Ranst, M. & Stevens, H. Rolling-circle amplification of viral DNA genomes using phi29 polymerase. *Trends in Microbiology* **17**, 205-211 (2009).
7. Baker, S.C. *et al.* The External RNA Controls Consortium: a progress report. *Nature Methods* **2**, 731-734 (2005).
8. Risso, D., Ngai, J., Speed, T.P. & Dudoit, S. Normalization of RNA-seq data using factor analysis of control genes or samples. *Nature Biotechnology* **32**, 896-902 (2014).
9. Robinson, M.D. & Oshlack, A. A scaling normalization method for differential expression analysis of RNA-seq data. *Genome Biol* **11**, R25 (2010).
10. Grün, D. & van Oudenaarden, A. Design and Analysis of Single-Cell Sequencing Experiments. *Cell* **163**, 799-810 (2015).
11. Qing, T., Yu, Y., Du, T. & Shi, L. mRNA enrichment protocols determine the quantification characteristics of external RNA spike-in controls in RNA-Seq studies. *Science China Life Sciences* **56**, 134-142 (2013).
12. Moller, H.D., Parsons, L., Jorgensen, T.S., Botstein, D. & Regenberg, B. Extrachromosomal circular DNA is common in yeast. *Proc Natl Acad Sci U S A* **112**, E3114-22 (2015).
13. Møller, H.D., Ramos-Madrigal, J., Prada-Luengo, I., Gilbert, M.T.P. & Regenberg, B. Near-Random Distribution of Chromosome-Derived Circular DNA in the Condensed Genome of Pigeons and the Larger, More Repeat-Rich Human Genome. *Genome Biology and Evolution* **12**, 3762-3777 (2020).

Decision Letter, second revision:

3rd Feb 2023

Dear Dr. Henssen,

Thank you for submitting your revised manuscript "Parallel sequencing of extrachromosomal circular DNAs and transcriptomes in single cancer cells" (NG-TR59148R1). It has now been seen by Reviewers #3 and #4 and their comments are below. As before, they reviewed the paper together and uploaded the same report.

The reviewers find that the paper has improved in revision, and therefore we'll be happy in principle to publish it in Nature Genetics, pending minor revisions to satisfy the referees' final requests and to comply with our editorial and formatting guidelines.

Sincerely,

Safia Danovi
Editor
Nature Genetics

Reviewer #3 (Remarks to the Author):

The authors have included a statement that their method is a low-throughput in the discussion section and added the sentence that described the overall timeline of their experiments. Although the authors mentioned that they cannot provide accurate copy number estimation of the ecDNA, they do provide relative copy number estimation data that suggest the scEC&T-seq result is somewhat consistent with their FISH results. In this revision the authors have provided additional data and descriptions that addressed our remaining minor concerns. Therefore, at this stage we recommend acceptance of their work.

Reviewer #4 (Remarks to the Author):

The authors have included a statement that their method is a low-throughput in the discussion section and added the sentence that described the overall timeline of their experiments. Although the authors mentioned that they cannot provide accurate copy number estimation of the ecDNA, they do provide relative copy number estimation data that suggest the scEC&T-seq result is somewhat consistent with their FISH results. In this revision the authors have provided additional data and descriptions that

addressed our remaining minor concerns. Therefore, at this stage we recommend acceptance of their work.

Final Decision Letter:

28th Mar 2023

Dear Dr. Henssen,

I am delighted to say that your manuscript "Parallel sequencing of extrachromosomal circular DNAs and transcriptomes in single cancer cells" has been accepted for publication in an upcoming issue of Nature Genetics.

Your paper will be published online after we receive your corrections and will appear in print in the next available issue. You can find out your date of online publication by contacting the Nature Press Office (press@nature.com) after sending your e-proof corrections. Now is the time to inform your Public Relations or Press Office about your paper, as they might be interested in promoting its publication. This will allow them time to prepare an accurate and satisfactory press release. Include your manuscript tracking number (NG-TR59148R2) and the name of the journal, which they will need when they contact our Press Office.

Acceptance is conditional on the data in the manuscript not being published elsewhere, or announced in the print or electronic media, until the embargo/publication date. These restrictions are not intended to deter you from presenting your data at academic meetings and conferences, but any

enquiries from the media about papers not yet scheduled for publication should be referred to us.

Please note that *Nature Genetics* is a Transformative Journal (TJ). Authors may publish their research with us through the traditional subscription access route or make their paper immediately open access through payment of an article-processing charge (APC). Authors will not be required to make a final decision about access to their article until it has been accepted. [Find out more about Transformative Journals](https://www.springernature.com/gp/open-research/transformative-journals)

Authors may need to take specific actions to achieve [compliance with funder and institutional open access mandates](https://www.springernature.com/gp/open-research/funding/policy-compliance-faqs). If your research is supported by a funder that requires immediate open access (e.g. according to [Plan S principles](https://www.springernature.com/gp/open-research/plan-s-compliance)) then you should select the gold OA route, and we will direct you to the compliant route where possible. For authors selecting the subscription publication route, the journal's standard licensing terms will need to be accepted, including [self-archiving and license to publish](https://www.nature.com/nature-portfolio/editorial-policies/self-archiving-and-license-to-publish). Those licensing terms will supersede any other terms that the author or any third party may assert apply to any version of the manuscript.

Please note that Nature Portfolio offers an immediate open access option only for papers that were first submitted after 1 January, 2021.

If you have not already done so, we invite you to upload the step-by-step protocols used in this

manuscript to the Protocols Exchange, part of our on-line web resource, natureprotocols.com. If you complete the upload by the time you receive your manuscript proofs, we can insert links in your article that lead directly to the protocol details. Your protocol will be made freely available upon publication of your paper. By participating in natureprotocols.com, you are enabling researchers to more readily reproduce or adapt the methodology you use. Natureprotocols.com is fully searchable, providing your protocols and paper with increased utility and visibility. Please submit your protocol to <https://protocolexchange.researchsquare.com/>. After entering your nature.com username and password you will need to enter your manuscript number (NG-TR59148R2). Further information can be found at <https://www.nature.com/nature-portfolio/editorial-policies/reporting-standards#protocols>

Sincerely,

Safia Danovi
Editor
Nature Genetics